# Multimodal Prompt Optimization: Why Not Leverage Multiple Modalities for MLLMs

**Yumin Choi**[*1]    **Dongki Kim**[*1]    **Jinheon Baek**[†1]    **Sung Ju Hwang**[†1,2]
[1]KAIST    [2]DeepAuto.ai
{yuminchoi, cleverki, jinheon.baek, sungju.hwang}@kaist.ac.kr

## Abstract

Large Language Models (LLMs) have shown remarkable success, and their multimodal expansions (MLLMs) further unlock capabilities spanning images, videos, and other modalities beyond text. However, despite this shift, prompt optimization approaches, designed to reduce the burden of manual prompt crafting while maximizing performance, remain confined to text, ultimately limiting the full potential of MLLMs. Motivated by this gap, we introduce the new problem of *multimodal prompt optimization*, which expands the prior definition of prompt optimization to the multimodal space defined by the pairs of textual and non-textual prompts. To tackle this problem, we then propose the **M**ultimodal **P**rompt **O**ptimizer (**MPO**), a unified framework that not only performs the joint optimization of multimodal prompts through alignment-preserving updates but also guides the selection process of candidate prompts by leveraging earlier evaluations as priors in a Bayesian-based selection strategy. Through extensive experiments across diverse modalities that go beyond text, such as images, videos, and even molecules, we demonstrate that MPO outperforms leading text-only optimization methods, establishing multimodal prompt optimization as a crucial step to realizing the potential of MLLMs.

## 1 Introduction

Large Language Models (LLMs) have demonstrated outstanding capabilities across a diverse range of tasks and domains (OpenAI, 2024; Grattafiori et al., 2024; Yang et al., 2025). We note that a central factor in unlocking their full potential lies in the design of prompts, which directly influence model performance. However, crafting high-quality prompts is often a labor-intensive and iterative process that requires substantial human intervention. To address this limitation, the field of Automatic Prompt Optimization (APO) has emerged, whose goal is to automate the discovery of effective prompts (Pryzant et al., 2023; Ramnath et al., 2025; Cui et al., 2025a; Li et al., 2025). For example, Zhou et al. (2023) frames this challenge as an iterative search problem, where at each step, a set of new candidate prompts is generated or updated, evaluated on a target task, and the best-performing prompts are selected to guide the next round of generation.

Recently, on top of the LLMs, Multimodal Large Language Models (MLLMs) have been proposed, which process not only text but also images, videos, and other modalities (such as molecules) (Kim et al., 2025; Zheng et al., 2024; Song et al., 2025). Yet, despite these advances and their wide-ranging applications, existing prompt optimization methods remain restricted to the textual modality (Choi et al., 2025; Fernando et al., 2024; Cui et al., 2025b), and overlook the richer expressive capacity afforded by multimodal inputs (that text alone cannot capture). For instance, as illustrated in Figure 1, describing the distinct characteristics of a specific bird may require long and potentially ambiguous text, while a single image can convey the same information far more directly. By limiting optimization to text, existing methods are prone to generating less effective, suboptimal prompts that fail to fully exploit the multimodal space that MLLMs are inherently capable of leveraging.

Motivated by this limitation, we first define the novel problem of *multimodal prompt optimization*[1], which expands the prompt optimization space beyond text to incorporate multiple modalities. However, while this expanded space opens new opportunities, it also introduces a couple of challenges

---

[*]Equal Contribution; [†]Equal Advising; Code is available at https://github.com/Dozi01/MPO.
[1]We define a multimodal prompt as a pair of a discrete textual and non-textual prompts.

Figure 1: **Concept Figure.** (A) Existing prompt optimization approaches restrict the optimization to the textual space, leaving MLLMs underutilized by failing to provide rich contextual signals. (B) Our multimodal prompt optimization expands the optimization space into multimodality, allowing the discovery of salient multimodal context and fully leveraging the expressive capacity of MLLMs.

for automatic optimization. First, exploring the larger, combinatorial space of multimodal prompts requires a prompt update strategy that can efficiently navigate candidate prompts while maintaining cross-modal consistency. Furthermore, selecting promising candidates becomes substantially more difficult, as the enlarged search space makes optimal prompts increasingly sparse, given the need to account for both the effectiveness within each modality and the alignment across modalities, which, in turn, calls for evaluation strategies that are both efficient and accurate.

To address these challenges, we propose **Multimodal Prompt Optimizer (MPO)**, a unified framework for optimizing prompts across both the textual and non-textual modalities, which consists of the two key components: (i) alignment-preserving exploration and (ii) prior-inheritance-based selection. Specifically, for exploration, the proposed MPO jointly updates the textual prompt, as well as its associated non-textual counterparts by generating instructions to create (or revise) the non-textual components of the multimodal prompt (unlike prior approaches that refine only text), and notably, their updates are guided by the single semantic gradient (i.e., feedback) to ensure their alignment derived from the failure analysis of the current prompt. Moreover, these updates are further diversified through complementary operations, namely generation, editing, and mixing, to ensure the broad and expressive exploration of the multimodal prompt space. Then, building on this exploration with multiple candidate prompts updated, MPO leverages the prior-inherited Bayesian-UCB as a prompt selection strategy, which utilizes the performance score of parent prompts as a prior (unlike conventional approaches that treat each candidate independently), to reliably identify the high-performing prompts by biasing the selection process toward more promising regions of the multimodal space.

To validate MPO, we conduct extensive experiments benchmarking it against leading text-only optimization methods across 10 datasets, and our evaluation suite spans not only images and videos but also molecular structures, ensuring broad coverage of diverse modalities. Across all domains, MPO demonstrate consistent and significant performance gains, empirically confirming our core hypothesis: expanding the prompt search space into the multimodal domain is crucial to exploit the expanded capacity of MLLMs. Further analyses show the efficacy of MPO components: alignment-preserving exploration with complementary operators facilitates the discovery of optimal multimodal prompts by not only ensuring cross-modal consistency but also thoroughly probing the search space; and the prior-inherited Bayesian-UCB accurately and efficiently selects high-performing prompts, reducing evaluation budget by 42% compared with a prior-free baseline. These results highlight MPO as an effective framework for optimizing multimodal prompts, unlocking the full capabilities of MLLMs.

## 2 RELATED WORK

**Multimodal Large Language Models** The development of MLLMs has significantly extended the capabilities of traditional LLMs by enabling them to process and reason over diverse non-textual modalities, including images, videos, audio, and more (Liu et al., 2023; Chu et al., 2023; M. Bran et al., 2024). In particular, these models are typically trained through large-scale multimodal pre-training, which aligns modality-specific encoders (e.g., vision or audio) with LLM backbones, followed by post-training stages such as supervised fine-tuning and preference optimization to endow them with multimodal instruction-following abilities (Gemini, 2025; Bai et al., 2025; Zhu et al., 2025). Moreover, leveraging these capabilities, MLLMs have achieved strong performance on a broad range of tasks, from foundational ones such as classification and captioning, to domain-

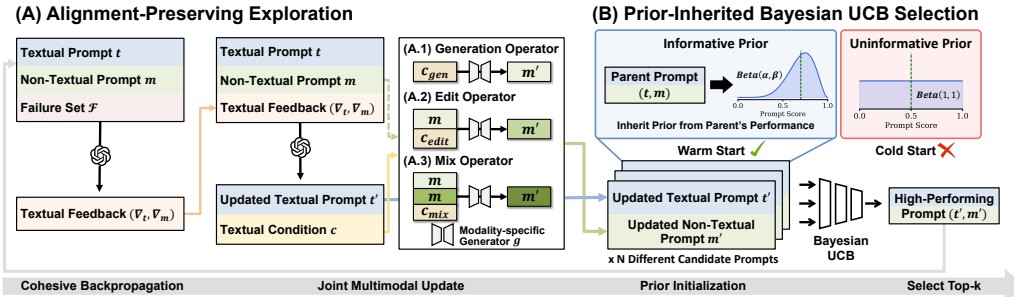

Figure 2: **Overview of MPO**, consisting of two components. (A) Alignment-preserving exploration analyzes a failure set to generate feedback, which is then used both to refine the textual prompt and to guide a modality-specific generator to create a new non-textual prompt with one of three operators. (B) Prior-Inherited Bayesian UCB Selection leverages the parent's performance as an informative prior, warm-starting the search to effectively identify high-performing prompts among candidates.

specific, high-stakes applications such as medical image question answering and pharmacological property prediction (Martin et al., 2019; Liu et al., 2021; Corbière et al., 2025; Huang et al., 2021).

**Automatic Prompt Optimization**   To reduce the burden of manual prompt engineering and systematically uncover the effective prompts, the field of Automatic Prompt Optimization (APO) has emerged. Existing works can be broadly categorized into two paradigms. The first is gradient-based optimization, which learns continuous embedding vectors (i.e., soft prompts) that are prepended to model inputs to steer behavior (Khattak et al., 2023; Zeng et al., 2024; Wang et al., 2024). Yet, while effective, they are computationally costly, yield uninterpretable numerical vectors, and are restricted to open-source models with accessible parameters. To overcome these drawbacks, gradient-free approaches have been proposed, which iteratively generate, evaluate, and refine candidate prompts using LLMs themselves (Zhou et al., 2023; Yang et al., 2024). Also, some recent works enhance this process by analyzing prompt failures to guide improvements (Khattab et al., 2024; Ye et al., 2024; Cui et al., 2025b; Yuksekgonul et al., 2025), while others borrow ideas from evolutionary algorithms (e.g., mutation and crossover) to explore the prompt space (Guo et al., 2024; Fernando et al., 2024). Despite this progress, current APO techniques are limited to text-only settings, restricting optimization to purely linguistic information. In contrast, our work expands prompt optimization into the multimodal domain, enabling the prompt discovery that fully exploits the capabilities of MLLMs.

**Instance-Specific Prompting and Optimization**   Distinct from task-level prompt optimization, another line of research focuses on instance-specific prompting strategies that operate at inference time to enhance reasoning on a per-query basis. For example, MM-CoT (Zhang et al., 2024b) guides the model to generate an intermediate textual rationale before producing the final answer. Also, other methods augment visual inputs with query-dependent signals, such as bounding boxes or points, to guide attention toward relevant regions of an image (Zhou et al., 2024; Jiang et al., 2024; Lin et al., 2024). Similar ideas have been explored in text-to-image and text-to-video generation, where prompts are crafted and refined to produce outputs more faithfully aligned with user intent (Mañas et al., 2024; Mo et al., 2024; Gao et al., 2025). However, these techniques are query-specific, designed to improve model performance for a single instance at a time. By contrast, APO pursues a different objective: discovering a single, reusable prompt that boosts performance across an entire task, and our work advances this paradigm by extending it into the multimodal domain.

## 3   METHODOLOGY: MULTIMODAL PROMPT OPTIMIZER

We present Multimodal Prompt Optimizer (MPO), composed of two modules: alignment-preserving exploration of multimodal prompt space and prompt selection with prior-inherited Bayesian UCB.

### 3.1   PROBLEM DEFINITION

We begin by formally describing MLLMs and then proposing a novel problem of multimodal prompt optimization, which redefines and expands the notion of existing prompt optimization beyond text.

**Multimodal Large Language Models**  Multimodal Large Language Models (MLLMs) extend the capabilities of LLMs by processing inputs that combine text with non-textual modalities. Formally, an MLLM can be represented as a parametric function $\texttt{MLLM} : (\mathcal{T} \cup \mathcal{M})^* \to \mathcal{T}$, where $\mathcal{T}$ denotes the textual input space, $\mathcal{M}$ denotes the non-textual input space, and $*$ denotes the Kleene Star (representing a finite sequence over the combined spaces). In other words, given a multimodal query $q$ and a prompt $p$ (each potentially containing both textual and non-textual components), the model generates a textual output $y = \texttt{MLLM}(p, q)$. It is worth noting that prior work on prompt optimization has generally restricted $p$ to a purely textual form ($p = t \in \mathcal{T}$), leaving the non-textual dimensions of $\mathcal{M}$ unused. This restriction underutilizes the expressive capacity of MLLMs and fails to provide richer contextual signals that are often crucial for real-world multimodal tasks (See Figure 1).

**Multimodal Prompt Optimization**  Building on the expanded space of MLLMs, we extend the notion of a prompt for optimization from text-only to multimodal. Specifically, we define a multimodal prompt as a pair $p = (t, m) \in \mathcal{T} \times \mathcal{M}$, where $t$ is the textual prompt and $m$ is the non-textual prompt. Then, given a task dataset $\mathcal{D}$ consisting of query–answer pairs $(q, a)$, the objective of multimodal prompt optimization is to discover the optimal prompt $(t^*, m^*)$ that maximizes performance:

$$(t^*, m^*) \;=\; \underset{(t,m)\in\mathcal{T}\times\mathcal{M}}{\operatorname{argmax}} \;\; \mathbb{E}_{(q,a)\sim\mathcal{D}}\Big[f\big(\texttt{MLLM}(t, m, q), a\big)\Big],$$

where $f$ is a function for a task-specific evaluation metric, such as accuracy or F1 scores.

Notably, compared to optimizing only textual prompts, the joint search space $\mathcal{T} \times \mathcal{M}$ introduces an entirely new axis of non-textual information, which in turn raises two fundamental challenges. First, multimodal prompts must maintain cross-modal consistency: textual and non-textual components should provide complementary, not conflicting signals; however, expanding to the combinatorial space greatly increases the risk of semantic misalignment. Second, the enlarged space amplifies the difficulty of candidate selection: high-quality prompts become sparse, and low-quality prompts dominate, making it harder to efficiently identify promising candidates. To overcome these, we now explain the proposed multimodal prompt optimizer, designed to navigate this enlarged space below.

## 3.2 ALIGNMENT-PRESERVING EXPLORATION OF MULTIMODAL PROMPT SPACE

The first challenge in multimodal prompt optimization lies in exploring the enlarged search space while preserving semantic consistency across modalities; thus, a naive approach that independently updates textual and non-textual components risks producing misaligned prompts, where one modality contradicts the other. To tackle this, we introduce an exploration framework that couples the update of textual and non-textual prompts while supporting diverse operations (Figure 2).

**Joint Optimization of Multimodal Prompt**  Our MPO jointly updates the textual and non-textual prompts to ensure that both evolve coherently, achieved through the following two mechanisms:

- *Cohesive Backpropagation.* We begin by identifying a failure set $\mathcal{F} = \{(q, a, y) \mid y \neq a\}$ for a multimodal prompt $p = (t, m)$. Instead of treating errors separately for text and non-textual inputs, we then generate a unified feedback $\nabla_p = (\nabla_t, \nabla_m) = \texttt{MLLM}(t, m; \mathcal{F})$, which encodes cross-modal weaknesses in textual form. By doing so, we obtain the single supervisory signal that guides both modalities simultaneously, mitigating the risk of overfitting updates to one modality.

- *Joint Multimodal Update.* Using the feedback, MPO jointly refines the textual prompt while deriving modality-specific conditions (in the textual form) that direct non-textual revisions. Specifically, the MLLM produces an updated textual prompt $t'$ and further a modality-specific condition $c$ describing how the non-textual prompt should adapt: $(t', c) = \texttt{MLLM}(t, m; \mathcal{F}, \nabla_p)$. The condition $c$ is then passed to modality-specific generators $g$ (such as text-to-image or text-to-molecule modules), which yield updated non-textual prompts $m' = g(c)$. This guarantees that updates to $m$ remain consistent with the revised textual prompt $t'$, rather than being optimized in isolation.

For the optimization process, we adopt the beam search. Specifically, at each iteration, we select top-$b$ best-performing multimdoal prompts $\{p_i = (t_i, m_i)\}_{i=1}^{b}$, and then apply the cohesive backpropagation and the joint multimodal update to generate new $b^2$ multimodal prompts $\{p'_j = (t'_j, m'_j)\}_{j=1}^{b^2}$ from the top-$b$ multimodal prompts $p$. We refer to $p$ and $p'$ as *parent* and *child* prompts, respectively.

**Exploration Operators**   Ensuring that generated outputs remain consistent with the guiding textual conditions is a necessary baseline, and effective optimization further requires $g$ that actively explores diverse regions of the multimodal space. To achieve this, we design three operators (namely, generation, edit, and mix), which systematically expand, refine, and recombine non-textual prompts.

- *Generation operator.* This operator explores entirely new non-textual prompts, e.g., novel spatial arrangements in visual inputs or unique substructures in molecules. Specifically, conditioned only on the generation signal $c_{\text{gen}}$, it creates a prompt from scratch without referencing prior candidates:

$$\boldsymbol{m}' = g(\boldsymbol{c}_{\text{gen}}, \varnothing), \text{ where } (\boldsymbol{c}_{\text{gen}}, \boldsymbol{t}') = \texttt{MLLM}(\boldsymbol{t}, \boldsymbol{m}; \nabla_{\boldsymbol{p}}, \mathcal{F}).$$

  By decoupling from past candidates, it explores unexplored regions and avoids local optima, especially in early stages (where initial prompts are unavailable) or when the candidate pool is biased.

- *Edit operator.* This operator performs fine-grained refinements of non-textual prompts (e.g., textures) while retaining useful structures from the prior prompt. Specifically, given the edit condition $c_{\text{edit}}$, the update is performed by conditioning on the prior non-textual prompt:

$$\boldsymbol{m}' = g(\boldsymbol{c}_{\text{edit}}, \{\boldsymbol{m}\}), \text{ where } (\boldsymbol{c}_{\text{edit}}, \boldsymbol{t}') = \texttt{MLLM}(\boldsymbol{t}, \boldsymbol{m}; \nabla_{\boldsymbol{p}}, \mathcal{F}).$$

  This enables targeted, incremental refinements, making it particularly effective when a prompt is already strong but requires adjustment on specific attributes rather than a complete redesign.

- *Mix operator.* This operator blends the complementary strengths of multiple multimodal prompts. Specifically, it first leverages feedback from multiple prompts to generate a mixing condition $c_{\text{mix}}$, which is then used by the generator to combine non-textual prompts as follows:

$$\boldsymbol{m}' = g(\boldsymbol{c}_{\text{mix}}, \{\boldsymbol{m}_i\}_{i=1}^{K}), \text{ where } (\boldsymbol{c}_{\text{mix}}, \boldsymbol{t}') = \texttt{MLLM}(\{\boldsymbol{t}_i, \boldsymbol{m}_i; \nabla_{\boldsymbol{p}_i}, \mathcal{F}_i\}_{i=1}^{K}).$$

  By synthesizing multiple candidates, it yields balanced compositions, avoids over-reliance on a single candidate, and enables exploration of intermediate solutions better than individual ones.

It is worth noting that, for the generation and edit operators, we randomly select one parent prompt from top-$b$ multimodal prompts of the previous iteration to generate a child prompt (i.e., $\boldsymbol{p} \to \boldsymbol{p}'$), while $K$ parent prompts are selected for the mix operator (i.e., $\{\boldsymbol{p}_1, ..., \boldsymbol{p}_K\} \to \boldsymbol{p}'$).

## 3.3   EFFECTIVE PROMPT SELECTION BY PRIOR-INHERITED BAYESIAN UCB

Another challenge in multimodal prompt optimization is to identify which candidates should be prioritized for evaluation and carried forward (Pryzant et al., 2023; Shi et al., 2024). Yet, this step is non-trivial with the enlarged multimodal space, since high-quality prompts become relatively sparse, and a large portion of the evaluation budget risks being wasted on low-potential candidates. Existing approaches typically adopt either (i) uniform allocation, where each candidate is evaluated equally regardless of its prior likelihood of success (Zhou et al., 2023; Cui et al., 2025b), or (ii) bandit-based allocation, such as UCB (Auer, 2002; Bouneffouf & Féraud, 2025; Li et al., 2026; Ashizawa et al., 2025), which adaptively balances exploration and exploitation. However, both paradigms suffer from an inefficient cold-start problem: newly generated prompts are treated as independent arms with no prior information, leading to unproductive evaluations in the early rounds.

**Parent-Child Correlation**   We address this cold-start inefficiency by introducing informative priors that warm-start the evaluation process. In particular, our hypothesis is that the performance of a parent prompt is positively correlated with that of its children. To test this, we analyze the optimization trajectory, measuring the correlation between the performance of parent prompts and the average performance of their children. As shown in Figure 3, we observe a strong positive correlation (Pearson's $r = 0.88$), providing concrete evidence that parent scores could serve as highly informative priors for estimating child performance.

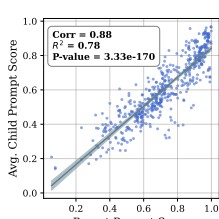

Figure 3: Correlation of parent and child scores.

**Prior-Inherited Bayesian UCB**   Motivated by this finding, we propose prior-inherited Bayesian UCB (Kaufmann et al., 2012), a selection strategy that initializes the score distribution of a new child prompt based on the posterior of its parent (rather than uniform). Specifically, we model the

expected score of each multimodal prompt $\boldsymbol{p}_i$ as a Beta distribution, $\mathrm{Beta}(\alpha_i, \beta_i)$, where $\alpha_i$ and $\beta_i$ correspond to (pseudo-) counts of successful and failure outcomes, respectively. Then, for a child prompt $\boldsymbol{p}_i$ originated from a parent prompt $\boldsymbol{p}_{\mathrm{par}(i)}$[2], we initialize its prior proportionally to the posterior mean performance of the parent $\hat{\mu}_{\mathrm{par}(i)}$, scaled by a prior strength hyperparameter $S > 0$, formalized as follows:

$$\alpha_i = \hat{\mu}_{\mathrm{par}(i)} \cdot S + 1, \ \ \beta_i = (1 - \hat{\mu}_{\mathrm{par}(i)}) \cdot S + 1, \quad \text{where} \quad \hat{\mu}_{\mathrm{par}(i)} = \frac{\alpha_{\mathrm{par}(i)}}{\alpha_{\mathrm{par}(i)} + \beta_{\mathrm{par}(i)}}. \quad (1)$$

This prior-inherited mechanism provides $S$ pseudo-observations to newly generated child prompts, effectively *warm-starting* the evaluation process. With a fixed total budget, it then proceeds iteratively: at each round, we select the prompt with the highest UCB score (an upper quantile of its Beta posterior), evaluate it on a small batch of data, and update its posterior parameters $\alpha_i$ and $\beta_i$. Once the budget is exhausted, the candidate prompt with the highest expected score is selected as the new parent for the next iteration of optimization. Please refer to Algorithm 2 for the complete procedure. The following proposition guarantees that our proposed selection strategy leverages an informative parent prior (better than random chance) to accelerate the selection of the best-promising prompt.

**Proposition 3.1.** *(Fewer Pulls via Prior-Inherited Bayesian UCB) With the prior of Equation 1, and if the prior is more informative than uniform ($\mathbb{E}_i\big[d(\mu_i, \hat{\mu}_{par(i)}) - d(\mu_i, \frac{1}{2})\big] \leq 0$), the best-arm identification cost of Bayesian UCB is nonincreasing, where $d(p, q)$ is the Bernoulli KL divergence.*

The proof and detailed analysis are provided in Appendix B. Intuitively, this guarantee demonstrates that informative parent priors accelerate the discovery of high-quality prompts by reducing wasted evaluations on low-potential candidates, which is particularly beneficial for multimodal prompt optimization, where the combinatorial search space is far larger than text-only settings. In other words, by rapidly eliminating unpromising candidates and reallocating the budget toward more promising regions, our method enables efficient exploration of the vast multimodal prompt landscape.

## 4 EXPERIMENTS

### 4.1 EXPERIMENTAL SETUP

**Datasets** We conduct an extensive evaluation on MPO across a diverse set of modalities, including images, videos, and molecules. For the image modality, we consider both image classification and visual question answering (VQA) tasks. Specifically, we use PlantVillage (Mohanty et al., 2016) for diseased leaf identification and CUB-200-2011 (Wah et al., 2011) for fine-grained bird classification; meanwhile, for VQA, we evaluate on SLAKE (Liu et al., 2021), RSVQA (Lobry et al., 2020), and DrivingVQA (Corbière et al., 2025), which cover radiology, remote sensing, and dynamic driving scenes, respectively. For the video modality, we evaluate on Drive&Act (Martin et al., 2019) for driver action recognition and VANE-Bench (Gani et al., 2025) for abnormality detection in video-based VQA. Finally, for the molecular modality, we include three different property prediction tasks from TDC (Huang et al., 2021), namely, Absorption (Hou et al., 2007; Ma et al., 2008; Broccatelli et al., 2011; Siramshetty et al., 2021), BBBP (Martins et al., 2012), and CYP inhibition tasks (Veith et al., 2009). Detailed configurations for each dataset are provided in Appendix A.1.

**Baselines** We benchmark MPO against both manually designed prompts and representative automatic prompt optimization methods. For manual prompting, we include **Human**, a simple hand-crafted prompt, **Chain-of-Thought (CoT)** (Wei et al., 2022), which uses the widely adopted phrase "Let's think step by step," and **Few-Shot**, which supplies in-context examples drawn from the training data. For automatic methods, we compare against leading LLM-based text-only optimizers, including **APE** (Zhou et al., 2023), **OPRO** (Yang et al., 2024), **EvoPrompt** (Guo et al., 2024), **PE2** (Ye et al., 2024), **ProTeGi** (Pryzant et al., 2023), and **SEE** (Cui et al., 2025b). Detailed descriptions of all baselines are provided in Appendix A.2.

**Implementation Details** For answer generation, we use Qwen2.5-VL (7B) (Bai et al., 2025) as the base model for image and video tasks, and Qwen3 (8B) (Yang et al., 2025) for molecular tasks. During optimization, GPT-4o mini (OpenAI, 2024) serves as the prompt optimizer, responsible for

---

[2]If a child prompt is generated by the mix operator, we average the posterior mean performance of multiple parent prompts.

Table 1: **Main Results.** Comparison of MPO with manual prompting, few-shot prompting, and text-only APO baselines on diverse benchmarks across image, video, and molecular modalities. Results are averaged over three independent runs. * denotes the average performance across multiple subtasks within the benchmark. Avg. denotes the average accuracy over all datasets except F1.

| | Image | | | | | Video | | Molecule | | | | | | |
| | PlantVillage* | CUB* | SLAKE* | DrivingVQA | RSVQA | Drive&Act | VANE. | Absorption* | | BBBP | | CYP Inhibit.* | | |
| Methods | Acc. | Acc. | Acc. | Acc. | Acc. | Acc. | Acc. | Acc. | F1 | Acc. | F1 | Acc. | F1 | Avg. |
|---|---|---|---|---|---|---|---|---|---|---|---|---|---|---|
| Human | 42.2 | 47.9 | 35.2 | 49.7 | 51.0 | 47.3 | 47.0 | 38.5 | 36.3 | 39.4 | 38.6 | 43.1 | 37.1 | 44.1 |
| CoT | 43.1 | 49.0 | 30.8 | 52.9 | 49.6 | 37.2 | 31.6 | 39.6 | 36.7 | 33.6 | 32.5 | 40.1 | 32.3 | 40.8 |
| 1-Shot | 39.7 | 54.7 | 31.4 | 54.5 | 48.5 | 50.4 | 62.4 | 37.8 | 35.7 | 36.1 | 34.8 | 56.2 | 48.3 | 47.2 |
| 3-Shot | 48.2 | 58.8 | 30.6 | 53.9 | 52.2 | 54.2 | 56.0 | 46.1 | 44.2 | 42.7 | 42.6 | 51.9 | 47.3 | 49.5 |
| 5-Shot | 46.5 | 58.1 | 28.0 | 45.9 | 49.2 | 54.3 | 61.4 | 48.1 | 45.5 | 49.3 | 49.3 | 52.0 | 47.0 | 49.3 |
| APE | 55.8 | 67.3 | 34.3 | 52.8 | 54.4 | 50.3 | 64.3 | 45.7 | 40.4 | 36.0 | 34.7 | 52.3 | 50.9 | 51.3 |
| OPRO | 54.1 | 59.7 | 33.9 | 52.7 | 51.0 | 46.4 | 51.0 | 37.6 | 35.4 | 39.2 | 38.3 | 43.0 | 37.1 | 46.9 |
| EvoPrompt | 56.1 | 59.6 | 34.8 | 52.9 | 50.5 | 46.7 | 56.5 | 48.2 | 46.5 | 38.7 | 37.7 | 51.1 | 49.7 | 49.5 |
| PE2 | 67.9 | 71.6 | 35.8 | 53.7 | 55.2 | 50.8 | 63.0 | 64.5 | 56.8 | 61.3 | 58.2 | 58.5 | 55.1 | 58.2 |
| ProTeGi | 64.4 | 70.0 | 35.4 | 54.4 | 54.2 | 53.0 | 65.5 | 71.1 | 58.2 | 72.1 | 65.7 | 59.8 | 57.0 | 60.0 |
| SEE | 69.0 | 71.6 | 35.0 | 52.2 | 53.4 | 51.7 | 57.9 | 71.4 | 60.0 | 67.0 | 62.3 | 61.4 | 56.7 | 59.1 |
| **MPO (Ours)** | **76.4** | **78.6** | **38.2** | **56.0** | **55.9** | **58.3** | **71.2** | **76.7** | **64.5** | **75.3** | **67.6** | **64.3** | **60.2** | **65.1** |

Table 2: **Generalizability results of MPO** across components with different backbones: (Top) base models; (Bottom Left) optimizer models; (Bottom Right) modality-specific generators.

| | Qwen2.5-VL (72B) | Gemma3 (12B) | InternVL-3.5 (14B) | GPT-4.1 nano |
|---|---|---|---|---|
| Human | 55.7 | 45.6 | 51.6 | 46.8 |
| 1-shot | 66.8 | 56.7 | 34.7 | 46.1 |
| 3-shot | 69.6 | 64.6 | 36.5 | 37.7 |
| 5-shot | 72.3 | 68.9 | 34.9 | 42.6 |
| ProTeGi | 74.1 | 68.2 | 71.9 | 61.0 |
| SEE | 73.6 | 68.1 | 70.8 | 61.6 |
| **MPO** | **80.4** | **73.1** | **73.2** | **65.9** |

| **Optimizer Model** | **SEE** | **MPO** |
|---|---|---|
| Qwen2.5-VL (7B) | 65.2 | **69.1** |
| Gemini 2.5 Flash | 68.2 | **74.8** |
| GPT-4o mini | 69.0 | **76.4** |
| GPT-4o | 69.2 | **78.0** |

| **T2I Generator** | **PlantVillage*** |
|---|---|
| SEE (Text-only) | 69.0 |
| SANA1.5 (1.6B) | 71.8 |
| Nano Banana | 72.9 |
| GPT-Image-Low | 76.4 |
| GPT-Image-Medium | 76.6 |

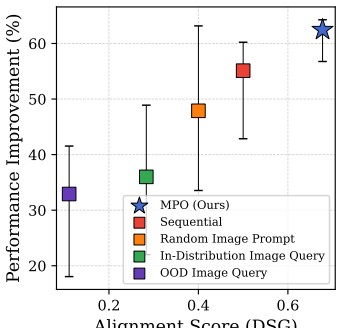

Figure 4: **Relationship between cross-modal alignment and performance gain**. We report median values alongside Q1 and Q3.

analyzing failures and refining multimodal prompts. For modality-specific generation, we employ GPT-Image (OpenAI, 2025) for images, Wan2.1 (1.3B) (Wan et al., 2025) for videos, and again GPT-4o mini for molecules. For the implementation of the iterative optimization loop, we use the beam search (Pryzant et al., 2023) with the beam size of $b = 3$ and the number of iterations of $T = 13$. Also, at each iteration (except the first), $b^2$ child prompts are produced by evenly applying the generation, edit, and mix operators, after which the top-$b$ prompts are selected via prior-inherited Bayesian-UCB. Meanwhile, in the first iteration, only the generation operator is used to initialize multimodal prompts, since no non-textual prompts exist yet. The complete optimization process is summarized in Algorithm 1. To ensure fairness, we keep the number of explored prompts consistent across all methods. In our case, each candidate prompt is allocated an evaluation budget of 100, and the prior strength for our prior inheritance is set to 10% of this budget ($S = 10$). Reported results are averaged over three independent runs. Please see Appendix A.3 for additional details.

## 4.2 EXPERIMENTAL RESULTS AND ANALYSES

**Main Results** As shown in Table 1, MPO consistently outperforms all baselines across image, video, and molecular domains, confirming its effectiveness in discovering prompts that more effectively harness the capabilities of MLLMs. Specifically, compared to existing text-only optimization methods, MPO achieves substantial gains, demonstrating that incorporating non-textual signals into prompts provides stronger contextual grounding and enhances task-specific reasoning. Moreover, MPO outperforms exemplar-based Few-Shot prompting, showing that it can capture richer cross-modal information and its underlying dependencies beyond simple query–answer demonstrations. In both image and video domains, MPO performs strongly on classification and QA tasks, underscoring its robustness across diverse real-world scenarios. Likewise, on molecular tasks, MPO surpasses all baselines, highlighting its effectiveness in highly specialized applications.

Table 3: Ablation on the contribution of each modality in the optimized multimodal prompt.

| Text | Image | PlantVillage* | CUB* |
|------|-------|---------------|------|
| Human | - | 42.2 | 47.9 |
| Human | MPO | 50.4 | 58.2 |
| MPO | - | 55.6 | 64.2 |
| MPO | MPO | **76.4** | **78.6** |

Table 4: Ablation on three exploration operators, utilizing each one of them individually.

| | Apple | Corn | Grape | Potato | Avg. |
|--|-------|------|-------|--------|------|
| SEE | 76.4 | 75.9 | 48.0 | 75.7 | 69.0 |
| Generation | 76.9 | 77.9 | 53.7 | 83.6 | 73.3 |
| Edit | 77.2 | 76.3 | 56.2 | 80.1 | 72.5 |
| Mix | 74.0 | 77.9 | 65.1 | 79.8 | 74.8 |
| MPO (Full) | **77.7** | **78.2** | **65.9** | **84.0** | **76.4** |

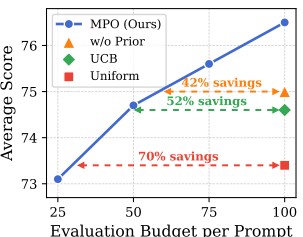

Figure 5: Efficiency comparison of selection strategies.

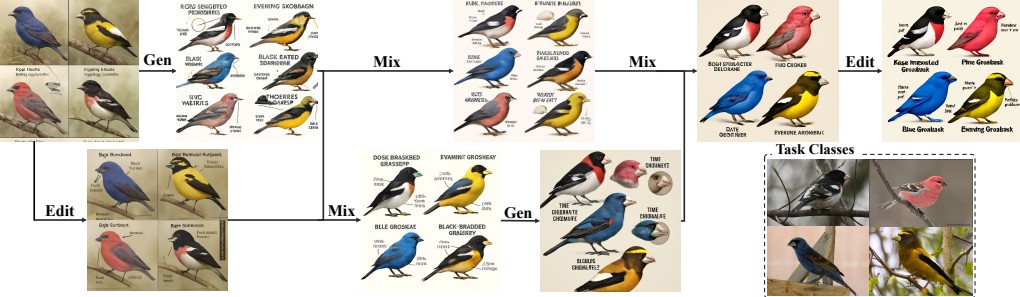

Figure 6: **Image prompt optimization process** of the best-performing multimodal prompt on a subtask (i.e., grosbeak species classification) of CUB. "Task Classes" box contains the examples of four species: Rose Breasted Grosbeak, Pine Grosbeak, Blue Grosbeak, and Evening Grosbeak.

**Generalizability to Diverse Backbone Models**   We further validate the generalizability of MPO by varying the backbone models used in each component, namely, base models, optimizer models, and modality-specific generators, and assessing its robustness under these variations. First, as shown in Table 2 (Top), MPO maintains strong performance across different architectures and exhibits even greater effectiveness as model size increases, for example, with Qwen2.5-VL (72B). Also, Table 2 (Bottom Left) further shows that MPO remains effective regardless of the optimizer model, surpassing state-of-the-art text-only methods (e.g., SEE) under diverse backbone models for optimization. Finally, Table 2 (Bottom Right) demonstrates that MPO generalizes well to modality-specific generators, including lightweight open-source models such as SANA1.5 (1.6B), where it continues to outperform textual optimization methods. These results highlight MPO as a broadly generalizable and robust framework, effective across a wide variety of base models and practical scenarios.

**Analysis on Cross-Modal Alignment**   Recall that MPO uses the alignment-preserving exploration to jointly refine textual and non-textual components of multimodal prompts, and we further analyze how this cross-modal alignment strategy contributes to performance gains. To isolate this effect, we consider four variants: (1) *Sequential*, where the textual prompt is optimized first and the non-textual prompt is refined afterward; (2) *Random Image Prompt*, where the image component is replaced with another optimized image prompt (i.e., not jointly optimized with the text); (3) *In-Distribution Image Query*, where it is replaced with an image sampled from the same task; and (4) *OOD Image Query*, where it is replaced with an image sampled from a different task. After that, we measure the relationship between performance gain over the Human baseline, as well as the DSG score[3] (Cho et al., 2024), used as a standard metric for measuring cross-modal alignment following prior work (Mañas et al., 2024). As shown in Figure 4, MPO achieves both the highest alignment score and the largest performance gains, followed by Sequential optimization and Random Image Prompt, while In-Distribution and OOD Image Query lag significantly behind. These results coAnfirm that stronger cross-modal alignment directly translates to better task performance, and that alignment-preserving updates (included in MPO) are crucial in promoting modality consistency.

**Ablation on Modality Contributions in Prompts**   To examine the contribution of each modality within optimized prompts, we ablate the textual and non-textual components from the final multi-

---

[3]DSG decomposes the textual description into atomic, dependency-aware queries and verifies each using an MLLM, enabling a precise assessment of whether the visual content reflects the intended textual details.

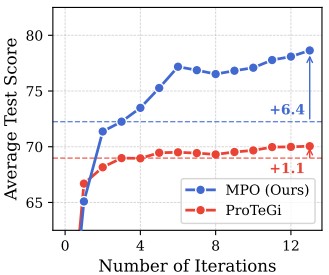 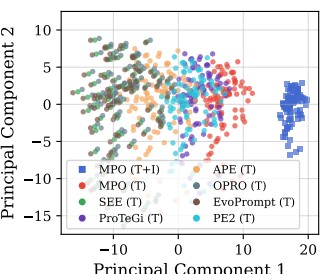 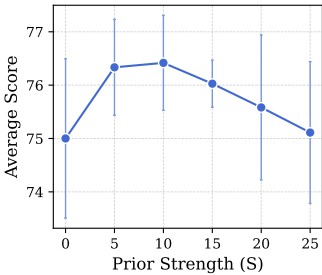

Figure 7: **Train Curve of MPO** compared to ProTeGi on CUB.  Figure 8: **Visualization of hidden states** in MLLMs by PCA.  Figure 9: **Analysis of the prior strength** ($S$) on performance.

modal prompt. As shown in Table 3, using only a single modality (either *MPO text without image* or *human text combined with MPO image*) already surpasses the Human baseline, confirming that both modalities independently provide useful signals. However, the full multimodal prompt yields substantially higher performance, demonstrating that the two modalities are not merely additive but mutually reinforcing, which underscores the importance of jointly leveraging textual and non-textual information to achieve performance gains beyond what either modality can deliver alone.

**Effect of Exploration Operators** To assess the contribution of the proposed exploration operators (such as generation, edit, and mix), we conduct both qualitative and ablation analyses. Qualitatively, as illustrated in Figure 6, we observe that each operator serves a distinct role: the *generation* operator introduces novel visual compositions, the *edit* operator fine-tunes local features such as textures or visual characters, and the *mix* operator blends broader attributes such as background or spatial layout. In addition to this, the ablation study in Table 4 further confirms their complementary effects: while each operator individually improves over the baseline, combining all three within MPO leads to the best performance. This demonstrates that the proposed operators jointly enable a more comprehensive exploration of the multimodal prompt space, facilitating the discovery of the optimal prompts. We observe a similar pattern in molecular prompt optimization, shown in Figure 14, with concrete examples of operator-driven updates (including textual conditions) provided in Table 8.

**Selection Strategies** We evaluate the effectiveness of our prior-inherited Bayesian UCB strategy for candidate prompt selection by comparing it against three alternatives: *Uniform*, which distributes the evaluation budget evenly across candidates; *UCB* (Auer, 2002), a standard bandit algorithm; and an ablated variant of ours *w/o Prior*. As shown in Figure 5, MPO achieves the same performance as the Uniform strategy while using only 30% of the evaluation budget, yielding a 70% reduction in resource cost. Moreover, MPO consistently outperforms both UCB and w/o Prior, reaching their performance levels with 52% and 42% less budget, respectively. These results confirm that the *warm start* enabled by prior inheritance is crucial for both efficiency and accuracy, allowing MPO to scale effectively over the enlarged multimodal search space and reliably identify high-quality prompts.

**Train Dynamics of MPO** To better understand how MPO improves over the course of optimization, we analyze its training dynamics in comparison to ProTeGi by tracking the test performance of the top-1 prompt on the CUB dataset. As shown in Figure 7, both methods improve during the early iterations; however, ProTeGi quickly plateaus after the third iteration, with only a marginal additional gain of 1.1 points. In contrast, MPO continues to improve steadily, ultimately achieving a much higher final score, including an additional 6.4-point gain beyond the third iteration. This comparison result highlights that MPO effectively overcomes the performance ceiling of text-only optimization methods by effectively navigating the multimodal prompt space, enabling it to escape local optima (imposed by the text-only strategy) and discover prompts closer to the global optimum.

**Hidden State Visualization** To gain deeper insight into why optimized multimodal prompts yield greater performance improvements than text-only prompts, we visualize the hidden state of MLLMs by averaging intermediate-layer embeddings, following Zhang et al. (2024a). As shown in Figure 8, hidden states obtained from text-only methods (including the text-only component of MPO) cluster together, suggesting that they guide the reasoning of MLLMs within a similar yet limited semantic space. In contrast, the full multimodal prompt from MPO shifts the hidden states into a distinct re-

gion, indicating that the non-textual component introduces information unavailable from text alone. In other words, the multimodal prompt alters the internal representation space of models, enabling richer reasoning pathways and ultimately leading to superior task performance.

**Analysis of Prior Strength**    Recall that in our prior-inherited selection strategy, the prior strength $S$ determines the number of pseudo-observations used to initialize the score distributions of child prompts, and we study its effect by varying $S$ and reporting the resulting performance. As shown in Figure 9, we first observe that a small $S$ under-utilizes the parent prior, resulting in weaker guidance and suboptimal performance. In contrast, an excessively large $S$ causes the model to over-rely on the parent prior, limiting its ability to adapt to the actual performance of child prompts. Consequently, the performance is maximized at an intermediate $S$, where inherited knowledge provides a strong warm start while still allowing sufficient flexibility to incorporate new observations.

**Qualitative Result**    We provide qualitative examples for the optimized multimodal prompts for the image modality in Table 9 of Appendix. From this, we observe that the optimized multimodal prompts consistently supply task-critical context in both textual and visual forms. Also, more importantly, the textual prompts explicitly instruct the model to leverage non-textual signals (e.g., Use the hybrid reference image for guidance), thereby unlocking the full multimodal capacity of MLLMs. Additional examples for the video and molecular modalities are presented in Tables 10, 11, and 12.

## 5    CONCLUSION

We introduced the novel problem of multimodal prompt optimization, extending the optimization space beyond text to fully leverage the capability of MLLMs. To tackle this, we proposed the Multimodal Prompt Optimizer (MPO), a unified framework that jointly refines textual and non-textual components through alignment-preserving exploration with multiple generation operations and efficiently identifies high-quality prompts via a prior-inherited Bayesian UCB strategy. Experiments across diverse modalities (including images, videos, and molecules) demonstrate that MPO consistently surpasses leading text-only prompt optimization methods, validating its efficacy in diverse real-world multimodal problems. We believe our work establishes multimodal prompt optimization as a key direction for advancing the use of MLLMs, moving beyond text-only prompting paradigms.

## ETHICS STATEMENT

Our study does not involve human subjects, personally identifiable data, or sensitive information. All experiments were conducted on public datasets and models under research-permissive licenses.

## REPRODUCIBILITY STATEMENT

We provide the public GitHub link to the code to ensure the reproducibility of our work. Additionally, we provide a detailed description of the experimental setup in Section 4.1. We further provide additional implementation details in Appendix A.3, the dataset configuration in Appendix A.1, the meta prompts to operationalize MPO in Appendix A.4, and the full algorithms in Appendix A.5.

## ACKNOWLEDGEMENT

This work was supported by the Institute for Information & communications Technology Planning & Evaluation (IITP) grant funded by the Korea government (MSIT) (RS-2019-II190075, Artificial Intelligence Graduate School Program (KAIST) and RS-2022-II220713, Meta-learning Applicable to Real-world Problems), the National Research Foundation of Korea (NRF) grant funded by the Korea government (MSIT) (No. RS-2023-00256259), the Korea Machine Learning Ledger Orchestration for Drug Discovery Project (K-MELLODDY) funded by the Ministry of Health & Welfare and Ministry of Science and ICT, Republic of Korea (No. RS-2024-00460870), the Basic Science Research Program through the National Research Foundation of Korea (NRF) funded by the Ministry of Education (RS-2024-00414751), the InnoCORE program of the Ministry of Science and ICT (No. N10250156), and the Center for Applied Research in Artificial Intelligence (CARAI) grant funded by DAPA and ADD (UD190031RD).

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

# A  ADDITIONAL EXPERIMENTAL DETAILS

## A.1  DETAILS ON DATASETS

We provide a detailed description of the datasets used in our experiments. To conduct a comprehensive evaluation, we compile a diverse set of benchmarks for classification and question-answering tasks across various modalities, including images, videos, and molecules. We use the official training/test splits where available, and if not, we create our own splits. For the image and video modality tasks, we sample 300 test examples, whereas for the molecule modality, we use the entire test set.

**PlantVillage**  The PlantVillage dataset (Mohanty et al., 2016) contains 54,306 images of plant leaves, spanning 38 disease categories across 14 crop species. To construct a focused, fine-grained classification task, we design subtasks by selecting four crop species, each having at least three distinct classes (e.g., one healthy and two or more diseases). This setup allows for a more controlled evaluation of the model's ability to identify specific plant diseases. Due to the lack of an official split, we split this subset using the 50/50 ratio for training and testing.

**CUB-200-2011**  The CUB-200-2011 dataset (Wah et al., 2011) is a standard benchmark for fine-grained bird species classification. To evaluate the capability of MLLMs in distinguishing between visually similar species, we group birds that share a common family name (e.g., "Hummingbird"), and select groups containing three or four distinct species to ensure a balanced level of difficulty, resulting in a total of 12 subtasks. Then, we divide the samples for each subtask using a 50/50 ratio, curating them to contain at least 80 instances for both training and test.

**SLAKE**  The SLAKE dataset (Liu et al., 2021) is an open-ended visual question answering benchmark tailored for the medical domain from various radiological modalities. To assess the performance of MLLMs across these different modalities, we partition the dataset into distinct subsets based on the modality, creating separate tasks for CT, MRI, and X-Ray images.

**DrivingVQA**  The DrivingVQA dataset (Corbière et al., 2025) is a closed-ended visual question answering benchmark with 3,931 multiple-choice questions based on real-world driving scenarios. To avoid ambiguity in the evaluation process, we filter the dataset to exclusively retain instances with a single correct answer, resulting in a final dataset of 2,039 training and 521 test instances.

**RSVQA**  We use the RSVQA dataset (Lobry et al., 2020) to evaluate performance on the open-ended visual question answering task for remote sensing images. Notably, the questions are designed to evaluate a model's understanding of various geospatial concepts, including land cover classification, object counting, and relational reasoning between objects. For our experiments, we utilize the low-resolution image set from the benchmark.

**Drive&Act**  For the video classification task, we use the Drive&Act dataset (Martin et al., 2019), which provides comprehensive labels for driver behaviors inside vehicles. We adhere to the official split of 6,642 training and 2,222 test instances, and preprocess the video clips by sampling frames at a rate of 1 frame per second (fps).

**VANE-Bench**  The VANE-Bench dataset (Gani et al., 2025) is a closed-ended question answering benchmark for video anomaly detection, whose samples (each with a 10-frame clip from synthetic or real-world videos) show various irregularities or distortions. We split the dataset into training and test sets using the 60/40 ratio, resulting in 293 training and 263 test instances.

**Absorption**  The Absorption task (Huang et al., 2021) is categorized into molecular property prediction, designed to evaluate a model's ability to estimate pharmacokinetic characteristics related to drug absorption. It is composed of four subtasks: PAMPA (Parallel Artificial Membrane Permeability Assay), HIA (Human Intestinal Absorption), Pgp (P-glycoprotein substrate classification), and Bioavailability, and we use the official random split.

**BBBP**  BBBP (Martins et al., 2012) is a molecular classification task to predict whether the given molecule can penetrate the blood-brain barrier (BBB), which is a highly selective system. We use the official random split from Huang et al. (2021), consisting of 1,453 train and 382 test examples.

**CYP Inhibit**  The CYP Inhibition task (Veith et al., 2009) involves classifying whether a molecule can inhibit Cytochrome P450 (CYP) enzymes, which play key roles in metabolism. It comprises five subtasks: inhibition of CYP 2C19, CYP 2D6, CYP 3A4, CYP 1A2, and CYP 2C9. We adopt the official random split provided in Huang et al. (2021).

## A.2  DETAILS ON BASELINES

This subsection details the baseline methods used in our experiments.

- **APE** (Zhou et al., 2023) generates candidate prompts by reverse-engineering instructions from examples and by paraphrasing existing prompts.
- **OPRO** (Yang et al., 2024) leverages the LLM as an optimizer, guiding it with pairs of prompts and their performance scores to generate progressively better instructions.
- **EvoPrompt** (Guo et al., 2024) utilizes an evolutionary algorithm where the LLM performs mutation and crossover operations on a population of prompts.
- **PE2** (Ye et al., 2024) focuses on optimizing the meta-prompt used to steer the LLM optimizer. It provides guidance through a structured template containing detailed task descriptions, context specification, and a step-by-step reasoning format.
- **ProTeGi** (Pryzant et al., 2023) simulates gradient descent for discrete prompts. It uses the LLM to generate natural language critiques based on prompt failures (termed "textual gradients"), and subsequently edits the prompt in the opposite semantic direction.
- **SEE** (Cui et al., 2025b) performs cohesive optimization of both the prompt instructions and the in-context examples. The method follows a four-phase process that strategically alternates between global exploration and local exploitation.

## A.3  ADDITIONAL IMPLEMENTATION DETAILS

In this subsection, we provide the additional implementation details in our experiments. Regarding model temperature, we use a temperature value of 0 for the base model to ensure consistency and 0.7 for the optimizer model to encourage the generation of diverse candidate prompts. The failure set size in the cohesive backpropagation process is fixed at 3. While the evaluation budget is generally set to 100, for CUB subtasks with fewer than 100 training samples, the budget for our MPO method is specifically set to one-third of the available instances. For modality-specific handling, we implement several strategies. In the video task, when the video query is part of the failure set, we sample three representative frames (first, middle, and last) from queries. In video generation, to mitigate the high complexity of video editing and mixing, we employ only the generation operator. We generate 5-second videos at 16 fps, then downsample them to 5 frames at 1 fps to construct the video prompt. For the molecule tasks, we represent chemical structures using the 1D representation (i.e., SMILES) and utilize GPT-4o mini for the molecule generator. Regarding optimization objectives, we use accuracy for image and video modalities, and F1 for the molecular modality to handle the class imbalance. Finally, to measure answer correctness, we adopt task-specific evaluation criteria: the final predefined label is extracted for standard classification, strict formatting rules are applied for binary and closed-ended QA tasks, and exact match is used for open-ended QA tasks. We select the best-performing prompts on the training set and report their performance on the test set. Our experiments are conducted on NVIDIA H100 80GB GPUs.

### A.4 META PROMPTS TO IMPLEMENT MPO

This subsection details the meta-prompts to instantiate MPO, which include a cohesive backpropagation prompt and three operator prompts (generation, edit, mix) for update. We provide the meta prompt from image modality as a representative example. The prompts for other modalities, such as video and molecule, are based on this structure, with minor, modality-specific wordings adjusted.

---

**Prompt for Cohesive Backpropagation**

You are a Prompt Failure Analysis Agent specialized in multimodal prompt optimization. Your task is to analyze the failure case of a Multimodal Large Language Model (MLLM) and identify the potential reasons in the prompt for the model's incorrect prediction. Based on the given input, output, and ground truth, analyze both the Text Prompt and the Image Prompt used in the task.

### Input Structure for MLLM:
- Text Prompt: A task-specific textual instruction for the MLLM.
- Image Prompt: A reference image that supports task understanding.
- Input Query: The actual target instance (text, image, or both) on which the MLLM must generate an answer.

### Prompts:
- Text Prompt : {text_prompt}
- Image Prompt : {modality_prompt}

### Wrong Examples:
{wrong_examples}

### Output Format:
Text Prompt Analysis:
- Identify missing information, vague instructions, or ambiguous wording that could have misled the model.
- Explain how weaknesses in the Text Prompt may have contributed to the wrong output.
- Suggest specific improvements (e.g., clearer task definition, additional constraints, better examples) to help the model produce the correct answer.

Image Prompt Analysis:
- If an image Prompt was used, analyze its effectiveness.
- Identify problems such as lack of clarity, poor composition, irrelevant details, or missing key features.
- If no image Prompt was used, suggest what kind of image (visual content, attributes, composition) would help correct the failure.

Figure 10: Meta Prompt for Cohesive Backpropagation in MPO.

---

**Prompt for Generation Operator**

You are a Prompt-Improvement Agent specializing in multimodal prompt optimization. Your task is to design improved prompts for both image generation and text instruction, aimed at enhancing the performance of Multimodal Large Language Model (MLLM).

### Input Structure for MLLM:
- Text Prompt: A task-specific textual instruction for the MLLM.
- Image Prompt: A reference image that supports task understanding.
- Input Query: The actual target instance (text, image, or both) on which the MLLM must generate an answer.

### Provided Material
- Text Prompt: {text_prompt}
- Image Prompt: {modality_prompt}
- Wrong Examples: {wrong_examples}
- Failure Analysis: {analysis}

### Your Task
Your task is to review the failure analysis carefully to understand the issues and create two improved prompts that directly address the issues in the failure analysis:
1. Image Generation Prompt
   - Write a detailed prompt for an image generator.
   - Enhance or redesign the reference image to resolve issues found in the analysis.
   - Ensure the image highlights critical visual features necessary for success.
   - If no reference image is provided, suggest an appropriate one based on the failure analysis.

2. Improved Text Prompt
   - Write a clear, concise, and unambiguous instruction for the MLLM.
   - Resolve ambiguities found in the failure analysis.
   - Elaborate on how the reference image should be interpreted.

### Output Format
<image_generation_prompt>{image_generation_prompt}</image_generation_prompt>
<improved_text_prompt>{improved_text_prompt}</improved_text_prompt>

Figure 11: Meta Prompt for Generation Operator in MPO.

```
Prompt for Edit Operator

You are a Prompt-Improvement Agent specializing in multimodal prompt optimization, with a focus on
prompt editing. Your task is to design improved prompts for both image editing and text instruction,
aimed at enhancing the performance of Multimodal Large Language Model (MLLM).

### Input Structure for MLLM:
- Text Prompt: A task-specific textual instruction for the MLLM.
- Image Prompt: A reference image that supports task understanding.
- Input Query: The actual target instance (text, image, or both) on which the MLLM must generate an
answer.

### Provided Material
- Text Prompt: {text_prompt}
- Image Prompt: {modality_prompt}
- Wrong Examples: {wrong_examples}
- Failure Analysis: {analysis}

### Your Task
Your task is to review the failure analysis carefully to understand the issues and create two improved
 prompts that directly address the issues in the failure analysis:
1. Image Editing Prompt:
   - Write a precise and context-aware prompt instructing the image editor to modify the given
reference image.
   - Specify which visual components (e.g., objects, colors, textures, lighting, perspective,
composition) should be added, removed, or replaced based on the failure analysis.
   - Clearly identify any undesirable visual elements that led to the failure.
   - Guide the editor on how to retain key features, proportions, or stylistic elements that are
critical to the intended outcome.

2. Improved Text Prompt
   - Write a clear, concise, and unambiguous instruction for the MLLM.
   - Resolve ambiguities found in the failure analysis.
   - Elaborate on how the reference image should be interpreted.

### Output Format
<image_edit_prompt>{image_edit_prompt}</image_edit_prompt>
<improved_text_prompt>{improved_text_prompt}</improved_text_prompt>
```

Figure 12: Meta Prompt for Edit Operator in MPO.

```
Prompt for Mix Operator

You are a Prompt-Improvement Agent specializing in multimodal prompt optimization, with a focus on
cross-prompt fusion. Your task is to create improved, mixed prompts for both image prompt and text
instruction, aimed at enhancing the performance of Multimodal Large Language Model (MLLM).

### Input Structure for MLLM:
- Text Prompt: A task-specific textual instruction for the MLLM.
- Image Prompt: A reference image that supports task understanding.
- Input Query: The actual target instance (text, image, or both) on which the MLLM must generate an
answer.

### Provided Material
#### Prompt A
- Text Prompt A: {text_prompt_A}
- Image Prompt A: {modality_prompt_A}
- Wrong Examples from Prompt A: {wrong_examples_A}
- Failure Analysis for Prompt A: {analysis_A}

#### Prompt B
- Text Prompt B: {text_prompt_B}
- Image Prompt B: {modality_prompt_B}
- Wrong Examples from Prompt B: {wrong_examples_B}
- Failure Analysis for Prompt B: {analysis_B}

### Your Task
Your task is to review the failure analysis carefully to understand the issues and create two improved
 prompts that directly address the issues in the failure analysis:
1. Image Mixing Prompt:
   - Write a guidance for the image generator to combine and improve both reference images.
   - Address visual issues identified in both failure analyses.
   - Guide the model to create a new hybrid image that merges key beneficial visual features from both
 references while mitigating their weaknesses.
   - Explicitly state which visual elements from each image should be retained, modified, or discarded
 to achieve task success.

2. Improved Text Prompt
   - Write a clear, concise, and unambiguous instruction for the MLLM.
   - Incorporate key visual or task-relevant features identified in both failure analysis.
   - Explain how the reference image should be used to assist the task.

### Output Format
<image_mixing_prompt>{image_mixing_prompt}</image_mixing_prompt>
<mixed_text_prompt>{mixed_text_prompt}</mixed_text_prompt>
```

Figure 13: Meta Prompt for Mix Operator in MPO.

## A.5 FULL ALGORITHM OF MPO

We provide the overall algorithm for MPO, with alignment-preserving exploration (including the operators) described in Algorithm 1 and the prior-inherited Bayesian UCB selection in Algorithm 2.

---

**Algorithm 1** MPO: Multimodal Prompt Optimizer

---

**Require:** Initial prompt $(\boldsymbol{t}_0, \varnothing)$, Number of iterations $T$, Beam size $b$
    Train dataset $\mathcal{D}_{tr}$, Metric function $f$

1:   $\boldsymbol{p} \leftarrow (\boldsymbol{t}_0, \varnothing)$, $\mathcal{P} \leftarrow \{\boldsymbol{p}\}$, $\mathcal{C} \leftarrow \varnothing$, $\hat{\mu} \leftarrow \mathbb{E}_{(\boldsymbol{q},\boldsymbol{a})\sim\mathcal{D}_{tr}}[f(\texttt{MLLM}(\boldsymbol{t}_0, \varnothing, \boldsymbol{q}), \boldsymbol{a})]$
2:   **for** $i = 1..b^2$ **do**
3:      $\mathcal{F}_{\boldsymbol{p}} \leftarrow \{(\boldsymbol{q}, \boldsymbol{a}, \boldsymbol{y}) \mid (\boldsymbol{q}, \boldsymbol{a})\sim\mathcal{D}_{tr},\ \boldsymbol{y} = \texttt{MLLM}(\boldsymbol{p}, \boldsymbol{q}),\ \boldsymbol{y} \neq \boldsymbol{a}\}$
4:      $\nabla_{\boldsymbol{p}} \leftarrow \texttt{MLLM.Feedback}(\boldsymbol{t}_0, \varnothing; \mathcal{F}_{\boldsymbol{p}})$
5:      $(\boldsymbol{t}', \boldsymbol{c}_{\text{gen}}) \leftarrow \texttt{MLLM.Generation}(\boldsymbol{t}_0, \varnothing; \nabla_{\boldsymbol{p}}, \mathcal{F}_{\boldsymbol{p}})$;   $\boldsymbol{m}' \leftarrow g(\boldsymbol{c}_{\text{gen}}, \varnothing)$
6:      $\mathcal{C} \leftarrow \mathcal{C} \cup \{(\boldsymbol{t}', \boldsymbol{m}')\}$
7:   **end for**
8:   $\mathcal{P} \leftarrow \texttt{BayesianUCBSelect}(\mathcal{P}, \mathcal{C}, b)$           ▷ Select $b$ prompts for next step
9:   **for** iter $= 1..T$ **do**
10:    $\mathcal{C} \leftarrow \varnothing$
11:    **for all** $\boldsymbol{p} = (\boldsymbol{t}, \boldsymbol{m}) \in \mathcal{P}$ **do**
12:      **for** $i = 1..b$ **do**
13:        $\mathcal{F}_{\boldsymbol{p}} \leftarrow \{(\boldsymbol{q}, \boldsymbol{a}, \boldsymbol{y}) \mid (\boldsymbol{q}, \boldsymbol{a})\sim\mathcal{D}_{tr},\ \boldsymbol{y} = \texttt{MLLM}(\boldsymbol{p}, \boldsymbol{q}),\ \boldsymbol{y} \neq \boldsymbol{a}\}$
14:        $\nabla_{\boldsymbol{p}} \leftarrow \texttt{MLLM.Feedback}(\boldsymbol{t}, \boldsymbol{m}; \mathcal{F}_{\boldsymbol{p}})$         ▷ Cohesive backpropagation
15:        op $\leftarrow \texttt{RandomSample}(\{\text{generation}, \text{edit}, \text{mix}\})$     ▷ Joint multimodal update
16:        **if** op $=$ generation **then**
17:          $(\boldsymbol{t}', \boldsymbol{c}_{\text{gen}}) \leftarrow \texttt{MLLM.Generation}(\boldsymbol{t}, \boldsymbol{m}; \nabla_{\boldsymbol{p}}, \mathcal{F}_{\boldsymbol{p}})$;   $\boldsymbol{m}' \leftarrow g(\boldsymbol{c}_{\text{gen}}, \varnothing)$
18:        **else if** op $=$ edit **then**
19:          $(\boldsymbol{t}', \boldsymbol{c}_{\text{edit}}) \leftarrow \texttt{MLLM.Edit}(\boldsymbol{t}, \boldsymbol{m}; \nabla_{\boldsymbol{p}}, \mathcal{F}_{\boldsymbol{p}})$;   $\boldsymbol{m}' \leftarrow g(\boldsymbol{c}_{\text{edit}}, \{\boldsymbol{m}\})$
20:        **else if** op $=$ mix **then**
21:          $\tilde{\boldsymbol{p}} \leftarrow \texttt{RandomSample}(\mathcal{P} \setminus \{\boldsymbol{p}\})$
22:          $(\boldsymbol{t}', \boldsymbol{c}_{\text{mix}}) \leftarrow \texttt{MLLM.Mix}((\boldsymbol{t}, \boldsymbol{m}; \nabla_{\boldsymbol{p}}, \mathcal{F}_{\boldsymbol{p}}), (\tilde{\boldsymbol{t}}, \tilde{\boldsymbol{m}}; \nabla_{\tilde{\boldsymbol{p}}}, \mathcal{F}_{\tilde{\boldsymbol{p}}}))$; $\boldsymbol{m}' \leftarrow g(\boldsymbol{c}_{\text{mix}}, \{\boldsymbol{m}, \tilde{\boldsymbol{m}}\})$
23:        **end if**
24:        $\mathcal{C} \leftarrow \mathcal{C} \cup \{(\boldsymbol{t}', \boldsymbol{m}')\}$
25:      **end for**
26:    **end for**
27:    $\mathcal{P} \leftarrow \texttt{BayesianUCBSelect}(\mathcal{P}, \mathcal{C}, b)$         ▷ Select $b$ prompts for next step
28:   **end for**
29:   **return** $\boldsymbol{p}^* \equiv (\boldsymbol{t}^*, \boldsymbol{m}^*)$ where $\boldsymbol{p}^* = \arg\max_{\boldsymbol{p}\in\mathcal{P}} \hat{\mu}_{\boldsymbol{p}}$,    $\hat{\mu}_{\boldsymbol{p}} = \frac{\alpha_{\boldsymbol{p}}}{\alpha_{\boldsymbol{p}}+\beta_{\boldsymbol{p}}}$

---

**Algorithm 2** Prior-Inherited Bayesian UCB Selection

---

**Require:** Parent prompts $\mathcal{P}$, A set of $k$ child prompts $\mathcal{C} = \{\boldsymbol{p}_i\}_{i=1}^k$, Beam size $b$
    Parent's performance $\{\hat{\mu}_{\text{par}(i)}\}_{i=1}^k$, Train dataset $\mathcal{D}_{tr}$, Metric function $f$, Batch size $B$
    Total evaluation budget $N$, Prior strength $S$, Exploration parameter $c$

1:   **Initialize** Beta priors for each child prompt $p_i \in \mathcal{P}$:
2:   **for** $i = 1, \ldots, k$ **do**
3:      $\alpha_i \leftarrow \hat{\mu}_{\text{par}(i)} \cdot S + 1$,    $\beta_i \leftarrow (1 - \hat{\mu}_{\text{par}(i)}) \cdot S + 1$          ▷ Inherit prior from parent
4:   **end for**
5:   **for** $t = 1, 2, \ldots, (N/B)$ **do**
6:      $q_t \leftarrow 1 - \frac{1}{t(\log N)^c}$
7:      $j \leftarrow \arg\max_{i\in\{1,..,k\}} \texttt{BetaQuantile}(q_t; \alpha_i, \beta_i)$      ▷ Choose prompt with highest UCB
8:      $\mathcal{D}_{mini} \leftarrow \texttt{Sample}(\mathcal{D}_{tr}, B)$
9:      $s_t \leftarrow \mathbb{E}_{(\boldsymbol{q},\boldsymbol{a})\sim\mathcal{D}_{mini}}[f(\texttt{MLLM}(\boldsymbol{t}, \boldsymbol{m}, \boldsymbol{q}), \boldsymbol{a})]$         ▷ Evaluate on small data batch
10:    $\alpha_j \leftarrow \alpha_j + s_t \cdot B$,    $\beta_j \leftarrow \beta_j + (1 - s_t) \cdot B$           ▷ Update posterior
11:   **end for**
12:   **Return** top-$b$ prompts from $\mathcal{P} \cup \mathcal{C}$ sorted by posterior mean $\hat{\mu}_i = \frac{\alpha_i}{\alpha_i+\beta_i}$

---

# B    THEORETICAL ANALYSIS ON PRIOR-INHERITED BAYESIAN UCB

In this section, we provide the proof for Proposition 3.1, starting with the formal problem setting.

**Setting.**    Let each arm $i \in \{1, \ldots, k\}$ have an unknown Bernoulli mean reward $\mu_i \in (0, 1)$ and let $i^\star \in \arg\max_i \mu_i$ be an optimal arm. Write the suboptimality gap as $\Delta_i = \mu_{i^\star} - \mu_i > 0$ for $i \neq i^\star$. As shown in algorithm 2, the Bayesian UCB algorithm maintains a Beta posterior distribution for each arm's mean reward. At each round $t$, Bayesian UCB selects the arm with the highest upper posterior quantile, $q_t$, observes the resulting Bernoulli reward, and updates the corresponding posterior.

**Prior inheritance**    For each child arm $i$, we initialize a Beta prior using the parent's posterior mean $\hat{\mu}_{\mathrm{par}(i)} \in (0, 1)$ and a pseudo-count $S > 0$:

$$\alpha_{0,i} = \hat{\mu}_{\mathrm{par}(i)} \, S + 1, \qquad \beta_{0,i} = (1 - \hat{\mu}_{\mathrm{par}(i)}) \, S + 1. \tag{2}$$

For comparison, a uninformative (or uniform) prior is $\mathrm{Beta}(1, 1)$. After $N_i(t)$ pulls with $X_i(t)$ successes by time $t$, the posterior parameters are $\alpha_i(t) = \alpha_{0,i} + X_i(t)$ and $\beta_i(t) = \beta_{0,i} + N_i(t) - X_i(t)$. Denote the posterior mean $\hat{\mu}_{t,i} = \alpha_i(t)/(\alpha_i(t) + \beta_i(t))$, the upper quantile $q_{t,i} = \mathrm{BetaQuantile}(q_t; \alpha_i(t), \beta_i(t))$, and the lower quantile $\ell_{t,i} = \mathrm{BetaQuantile}(1 - q_t; \alpha_i(t), \beta_i(t))$.

**Average KL-closeness assumption.**    Our analysis relies on the assumption that a parent's posterior provides a useful inductive bias for its children. We formalize this concept using the Kullback-Leibler (KL) divergence for Bernoulli distributions, defined as $d(p, q) = p \log \frac{p}{q} + (1 - p) \log \frac{1-p}{1-q}$. Let $\mathcal{I}$ be the population of child arms produced during optimization. We assume the parent estimate is, *on average over children*, KL-closer to the truth than the mean of the uninformative prior:

$$\mathbb{E}_{i \sim \mathcal{I}} \big[ d\big(\mu_i, \hat{\mu}_{\mathrm{par}(i)}\big) - d\big(\mu_i, \tfrac{1}{2}\big) \big] \leq -\gamma \quad \text{for some } \gamma > 0. \tag{3}$$

The assumption is empirically supported by the strong positive correlation observed between parent and child scores (Figure 3).

## B.1    TWO AUXILIARY LEMMAS

**Lemma B.1** (Pseudo-counts shrink one-sided credible widths). *There exists a universal constant $c > 0$ such that for all $t \geq 2$ and all arms $i$,*

$$q_{t,i} - \hat{\mu}_{t,i} \leq c \sqrt{\frac{\log t}{N_i(t) + S}}, \qquad \hat{\mu}_{t,i} - \ell_{t,i} \leq c \sqrt{\frac{\log t}{N_i(t) + S}}. \tag{4}$$

*The key implication is that the credible interval width scales with $1/\sqrt{N_i(t) + S}$ rather than $1/\sqrt{N_i(t)}$. Thus, the prior strength $S$ acts as an additive effective sample size, shrinking the interval as if we had $S$ additional observations.*

*proof sketch.* The proof relies on standard concentration bounds for Beta posteriors. The conjugacy of the Beta-Binomial model makes the posterior tractable, allowing for the specific application of a Chernoff tail bound. For any $\varepsilon \in (0, 1)$, the probability that the upper posterior quantile underestimates the true mean by at least

$$\mathbb{P}\{ q_{t,i} \leq \mu_i - \varepsilon \} \lesssim \exp\big( - (N_i(t) + S) \, d(\mu_i - \varepsilon, \mu_i) \big), \tag{5}$$

with a symmetric bound holding for the lower quantile $\ell_{t,i}$. The result is obtained by using the approximation $d(\mu_i - \varepsilon, \mu_i) \gtrsim \varepsilon^2$ for small $\varepsilon$ and selecting the quantile level $1 - q_t = \Theta(1/t)$ yields the stated $\sqrt{\log t/(N_i(t) + S)}$ bounds. $\qquad\square$

**Lemma B.2** (Effect of Informative Priors on Posterior Quantiles). *Under equation 3, for any fixed real counts $(n, s)$ with $s \sim \mathrm{Binom}(n, \mu_i)$, the posterior under prior inheritance $\mathrm{Beta}(\alpha_{0,i}+s, \beta_{0,i}+n-s)$ is, on average over $i \sim \mathcal{I}$, better centered around $\mu_i$ than the posterior under uninformative prior $\mathrm{Beta}(1 + s, 1 + n - s)$. Consequently,*

$$\mathbb{E}_{i \sim \mathcal{I}} \big[ \ell_{t,i^\star}^{(\mathrm{par})} - \ell_{t,i^\star}^{(\mathrm{unif})} \big] \geq 0, \qquad \mathbb{E}_{i \sim \mathcal{I}} \big[ q_{t,i}^{(\mathrm{par})} - q_{t,i}^{(\mathrm{unif})} \big] \leq 0 \quad (i \neq i^\star), \tag{6}$$

*with strict inequalities whenever $\gamma > 0$ and $S > 0$.*

*proof sketch.* The posterior mean under prior inheritance is $\hat{\mu}_{n,i}^{(\text{par})} = \frac{S\hat{\mu}_{\text{par}(i)}+s+1}{S+n+2}$, while the posterior mean under uninformative prior is $\hat{\mu}_{n,i}^{(\text{unif})} = \frac{1+s}{n+2}$. Taking expectation over $s$ and then over $i \sim \mathcal{I}$ yields convex combinations of $\mu_i$ with $\hat{\mu}_{\text{par}(i)}$ versus $1/2$. Our KL-closeness assumption equation 3 directly implies that the posterior mean under prior inheritance is, in expectation, a better estimate of $\mu_i$. Since the posterior quantiles are centered around this mean, Lemma B.1 ensures that an improvement in the mean's centering translates to the stated shifts in the quantiles, holding in expectation. $\qquad\square$

## B.2 Sufficient Condition for Optimal Arm Identification

For the algorithm to correctly identify the optimal arm $i^\star$ by the final round $T$, a sufficient condition is that the credible intervals for the optimal and suboptimal arms are well-separated. Formally, this occurs if the lower quantile of the optimal arm exceeds the upper quantile of every suboptimal arm. This is the separation event:

$$\ell_{T,i^\star} > \max_{i \neq i^\star} q_{T,i}. \tag{7}$$

If this separation event fails, it implies that for some suboptimal arm $i$, the credible intervals overlap. This allows us to bound the suboptimality gap $\Delta_i$ by the sum of the one-sided credible widths:

$$\Delta_i \leq (\mu_{i^\star} - \ell_{T,i^\star}) + (q_{T,i} - \mu_i) \lesssim \sqrt{\frac{\log T}{N_{i^\star}(T) + S}} + \sqrt{\frac{\log T}{N_i(T) + S}}, \tag{8}$$

where the second inequality follows from Lemma B.1. This implies that to guarantee separation, the credible interval widths must be sufficiently small relative to the gap. Therefore, a deterministic sufficient condition for equation 7 is: there exists a universal constant $c' > 0$ such that,

$$\sqrt{\frac{\log T}{N_{i^\star}(T) + S}} + \sqrt{\frac{\log T}{N_i(T) + S}} < c' \Delta_i. \tag{9}$$

Crucially, combining this condition with the quantile shift from Lemma B.2 reveals the benefit of our approach. Because the prior inheritance yields better quantile estimates, the sample allocation required to satisfy equation 9 is achieved no later than with an uninformative prior.

## B.3 Proof of Proposition 3.1

*Proof.* The prior inheritance improves the performance of Bayesian UCB through two synergistic mechanisms:

**(i) Tighter credible intervals at fixed counts.** For any given allocation of pulls, the prior strength $S$ acts as an additive effective sample size. As established in Lemma B.1, this shrinks the credible interval widths by effectively replacing the sample size $N_i(T)$ with $N_i(T)+S$. This directly reduces the left-hand side of the deterministic condition equation 9, making it easier to satisfy.

**(ii) More efficient sample allocation.** The informative prior also leads to a better allocation of pull over time. Lemma B.2 shows that the quantiles are favorably shifted on average: the lower bound for the optimal arm $i^\star$ increases, while the upper bounds for suboptimal arms decrease. This improved estimation guides the UCB policy to allocate more pulls to $i^\star$ and waste fewer on suboptimal arms, particularly in the early stages. Consequently, in expectation:

$$\mathbb{E}\left[N_{i^\star}^{(\text{par},S)}(T)\right] \geq \mathbb{E}\left[N_{i^\star}^{(\text{unif})}(T)\right], \qquad \mathbb{E}\left[N_i^{(\text{par},S)}(T)\right] \leq \mathbb{E}\left[N_i^{(\text{unif})}(T)\right] \quad (i \neq i^\star), \tag{10}$$

with strict inequalities when the prior is strictly beneficial ($\gamma > 0$ and $S > 0$).

Together, these two mechanisms ensure that the separation condition equation 7 is met more efficiently. The tighter intervals (i) make the condition easier to satisfy for any given sample allocation, and the improved allocation strategy (ii) finds a sufficient allocation faster. As a result, for a sufficiently larger budget $T$, the total expected number of pulls on suboptimal arms is reduced:

$$\mathbb{E}\left[\sum_{i \neq i^\star} N_i^{(\text{par},S)}(T)\right] \leq \mathbb{E}\left[\sum_{i \neq i^\star} N_i^{(\text{unif})}(T)\right], \tag{11}$$

This is equivalent to stating that the expected cost of identifying the best arm is non-increasing, and strictly decreases whenever the average KL-closeness assumption holds. $\qquad\square$

# C  ADDITIONAL EXPERIMENTAL RESULTS AND ANALYSIS

## C.1  COMPARISON OF COMPUTATIONAL COSTS

Table 5: Comparison of the number of model requests (or calls) for MPO and other baselines.

| Methods | Base Model | Optimizer Model | Modality-Specific Generator | Avg. Performance |
|---|---|---|---|---|
| APE | 11.7k | 117 | N/A | 51.3 |
| ProTeGi | 11.7k | 234 | N/A | 60.0 |
| SEE | 11.7k | 153 | N/A | 59.1 |
| MPO (Ours) | 11.7k | 234 | 117 | **65.1** |

We analyze the number of model requests (or model calls) as a proxy for computational cost, and report the results in Table 5. First, the base model call is the same for all methods, as we fix the number of explored prompts and the evaluation budget. For the optimizer model calls, APE uses the one-step exploration (e.g., paraphrasing), requiring the number of calls to be equal to the generated candidates. ProTeGi and our MPO utilize a two-step process (e.g., feedback generation and refinement), requiring twice the number of calls. SEE combines both approaches and falls in between. Note that, although MPO incurs an additional computational cost by calling a modality-specific generator to explore non-textual prompts, this cost is manageable, as this process can utilize lightweight, open-source generators such as SANA1.5 (1.6B) to minimize the additional expense, while still outperforming text-only prompt optimization methods as validated in Table 2 (Bottom Right). In other words, despite the marginal increase in computation (which is also manageable), MPO achieves a substantial performance improvement unattainable by existing text-only optimization methods.

## C.2  FULL MAIN RESULTS

We provide the full results, including performance on individual subtasks. The results for the image modality are presented in Table 6, and for the molecule modality in Table 7.

Table 6: Full experimental results on image modality benchmarks, including subtasks, with all scores reported as the average accuracy over three independent experiments.

| | PlantVillage | | | | | CUB | | | | | | | | | | | | | SLAKE | | | | DrivingVQA | RSVQA |
|---|---|---|---|---|---|---|---|---|---|---|---|---|---|---|---|---|---|---|---|---|---|---|---|---|
| | Apple | Corn | Grape | Potato | Avg | hummingbird | albatross | bunting | jay | cuckoo | cormorant | swallow | blackbird | auklet | grosbeak | oriole | grebe | Avg | CT | MRI | X-Ray | Avg | DrivingVQA | RSVQA |
| Human | 47.4 | 40.5 | 29.1 | 51.7 | 42.2 | 50.4 | 42.8 | 74.0 | 90.4 | 12.2 | 35.0 | 50.0 | 32.5 | 31.8 | 68.3 | 40.1 | 46.9 | 47.9 | 35.7 | 30.0 | 39.9 | 35.2 | 49.7 | 51.0 |
| CoT | 57.0 | 35.6 | 34.9 | 44.8 | 43.1 | 49.6 | 39.0 | 80.6 | 81.6 | 30.1 | 39.1 | 49.4 | 44.1 | 34.3 | 56.7 | 32.8 | 50.3 | 49.0 | 31.5 | 27.9 | 33.1 | 30.8 | 52.9 | 49.6 |
| 1-shot | 48.6 | 35.4 | 27.1 | 47.8 | 39.7 | 56.3 | 49.6 | 68.2 | 87.0 | 48.4 | 44.0 | 33.6 | 69.3 | 32.3 | 72.5 | 46.8 | 47.8 | 54.7 | 32.6 | 22.8 | 38.9 | 31.4 | 54.5 | 48.5 |
| 3-shot | 72.2 | 37.5 | 27.5 | 55.4 | 48.2 | 62.6 | 36.8 | 74.0 | 92.0 | 55.7 | 50.2 | 51.4 | 53.1 | 35.9 | 80.6 | 45.9 | 66.9 | 58.8 | 29.7 | 21.5 | 40.6 | 30.6 | 53.9 | 52.2 |
| 5-shot | 69.8 | 38.8 | 23.2 | 54.3 | 46.5 | 67.0 | 46.2 | 78.3 | 95.1 | 56.9 | 48.5 | 53.4 | 36.8 | 45.5 | 71.7 | 46.8 | 51.4 | 58.1 | 26.3 | 16.2 | **41.3** | 28.0 | 45.9 | 49.2 |
| APE | 70.7 | 66.0 | 33.8 | 52.9 | 55.8 | 80.4 | 56.4 | 89.2 | 96.9 | 46.1 | 56.0 | 54.4 | 80.0 | 41.9 | 87.5 | 45.7 | 73.1 | 67.3 | 35.9 | 28.9 | 38.1 | 34.3 | 52.8 | 54.4 |
| OPRO | 68.2 | 63.1 | 31.2 | 53.9 | 54.1 | 57.4 | 47.7 | 87.6 | 90.2 | 34.6 | 53.9 | 56.7 | 75.1 | 34.3 | 77.8 | 45.1 | 55.6 | 59.7 | 35.2 | 28.0 | 38.3 | 33.9 | 52.7 | 51.0 |
| EvoPrompt | 70.9 | 65.7 | 32.8 | 55.1 | 56.1 | 61.3 | 41.5 | 90.5 | 87.9 | 41.3 | 45.3 | 50.6 | 62.6 | 34.3 | 88.3 | 44.3 | 67.2 | 59.6 | 35.2 | 29.9 | 39.3 | 34.8 | 52.9 | 50.5 |
| PE2 | 74.0 | 74.8 | 43.7 | 79.2 | 67.9 | 78.5 | 60.6 | **94.6** | 94.6 | 46.8 | 54.3 | 67.2 | **89.6** | 45.5 | **98.9** | 53.2 | 75.8 | 71.6 | **36.8** | 31.9 | 38.9 | 35.8 | 53.7 | 55.2 |
| ProTeGi | 75.4 | 71.0 | 38.4 | 72.6 | 64.4 | **83.3** | 51.9 | 91.5 | **97.7** | 60.6 | 53.5 | 62.5 | 81.4 | 42.4 | 98.6 | 48.5 | 68.6 | 70.0 | 36.2 | 33.2 | 36.9 | 35.4 | 54.4 | 54.2 |
| SEE | 76.4 | 75.9 | 48.0 | 75.7 | 69.0 | 78.9 | 56.8 | 93.4 | 95.0 | 61.0 | **60.5** | 64.7 | 86.7 | 47.0 | 98.1 | 48.2 | 69.2 | 71.6 | 36.3 | 31.8 | 36.9 | 35.0 | 52.2 | 53.4 |
| MPO | **77.7** | **78.2** | **65.9** | **84.0** | **76.4** | 82.2 | **61.4** | **94.6** | 97.3 | **68.3** | 58.4 | **71.1** | 85.5 | **73.7** | 98.6 | **68.4** | **84.2** | **78.6** | 36.1 | **37.5** | 41.0 | **38.2** | **56.0** | **55.9** |

Table 7: Full experimental results on molecule modality benchmarks, including subtasks, with all scores reported as the average accuracy over three independent experiments.

| | Absorption | | | | | | | | | | BBBP | | CYP Inhibition | | | | | | | | | | | |
|---|---|---|---|---|---|---|---|---|---|---|---|---|---|---|---|---|---|---|---|---|---|---|---|---|
| | PAMPA | | HIA | | Pgp | | Bioavail. | | Avg | | BBBP | | CYP 2C19 | | CYP 2D6 | | CYP 3A4 | | CYP 1A2 | | CYP 2C9 | | Avg | |
| | Acc | F1 | Acc | F1 | Acc | F1 | Acc | F1 | Acc | F1 | Acc | F1 | Acc | F1 | Acc | F1 | Acc | F1 | Acc | F1 | Acc | F1 | Acc | F1 |
| Human | 18.2 | 16.8 | 41.1 | 40.7 | 55.6 | 49.4 | 39.1 | 38.2 | 38.5 | 36.3 | 39.4 | 38.6 | 52.6 | 46.2 | 25.8 | 25.1 | 43.6 | 33.5 | 52.9 | 43.7 | 40.5 | 37.1 | 43.1 | 37.1 |
| CoT | 30.2 | 31.1 | 40.2 | 40.8 | 51.3 | 36.1 | 36.7 | 38.9 | 39.6 | 36.7 | 33.6 | 32.5 | 48.7 | 39.1 | 22.8 | 21.5 | 42.8 | 32.7 | 50.0 | 38.7 | 36.2 | 29.5 | 40.1 | 32.3 |
| 1-shot | 16.0 | 14.0 | 40.2 | 39.7 | 58.6 | 55.4 | 36.2 | 33.6 | 37.8 | 35.7 | 36.1 | 34.8 | 56.8 | 50.9 | 61.4 | 37.6 | 54.9 | 52.5 | 56.3 | 50.4 | 51.8 | 50.4 | 56.2 | 48.3 |
| 3-shot | 23.3 | 22.7 | 56.9 | 52.7 | 58.2 | 56.0 | 46.1 | 45.6 | 46.1 | 44.2 | 42.7 | 42.6 | 53.1 | 48.1 | 48.5 | 40.0 | 47.1 | 40.3 | 60.4 | 58.6 | 50.7 | 49.2 | 51.9 | 47.3 |
| 5-shot | 23.3 | 23.1 | 66.7 | 59.2 | 58.8 | 56.3 | 43.8 | 43.5 | 48.1 | 45.5 | 49.3 | 49.3 | 55.7 | 53.1 | 44.2 | 38.3 | 48.3 | 42.8 | 55.3 | 48.3 | 56.4 | 52.7 | 52.0 | 47.0 |
| APE | 17.7 | 16.2 | 73.6 | 55.2 | 52.4 | 51.7 | 39.1 | 38.6 | 45.7 | 40.4 | 36.0 | 34.7 | 54.3 | 53.6 | 49.4 | 44.7 | 49.3 | 49.1 | 56.2 | 56.2 | 52.3 | 50.9 | 52.3 | 50.9 |
| OPRO | 18.0 | 16.6 | 40.2 | 39.9 | 56.3 | 50.0 | 35.9 | 34.9 | 37.6 | 35.4 | 39.2 | 38.3 | 52.6 | 46.3 | 25.7 | 25.0 | 43.5 | 33.4 | 52.9 | 43.7 | 40.5 | 37.1 | 43.0 | 37.1 |
| EvoPrompt | 36.4 | 35.8 | 55.8 | 51.8 | 52.4 | 50.6 | 48.4 | 47.6 | 48.2 | 46.5 | 38.7 | 37.7 | 52.8 | 52.1 | 46.3 | 42.8 | 48.8 | 46.9 | 57.7 | 57.2 | 49.8 | 49.4 | 51.1 | 49.7 |
| PE2 | 52.7 | 45.9 | 82.5 | 65.8 | 63.1 | 62.0 | 59.6 | 53.3 | 64.5 | 56.8 | 61.3 | 58.2 | 57.6 | 57.4 | 57.9 | 46.1 | 58.0 | 56.7 | 60.6 | 60.3 | 58.6 | 55.1 | 58.5 | 55.1 |
| ProTeGi | 74.8 | 54.1 | 84.5 | 64.4 | 59.9 | 59.5 | 65.1 | 54.8 | 71.1 | 58.2 | 72.1 | 65.7 | 58.8 | 58.3 | 57.6 | 48.6 | 60.3 | 57.2 | 61.7 | 61.5 | 60.4 | 59.3 | 59.8 | 57.0 |
| SEE | 68.8 | 52.5 | 85.1 | 69.7 | 65.4 | 65.1 | 66.4 | 52.6 | 71.4 | 60.0 | 67.0 | 62.3 | 56.4 | 56.4 | **70.7** | 51.1 | 57.7 | 57.7 | 62.1 | 61.7 | 59.8 | 56.8 | 61.4 | 56.7 |
| MPO | **78.6** | **56.1** | **89.1** | **76.3** | **71.0** | **70.6** | **68.2** | **55.1** | **76.7** | **64.5** | **75.3** | **67.6** | **60.2** | **59.2** | 67.6 | **51.9** | **64.2** | **63.5** | **64.1** | **63.6** | **65.4** | **62.5** | **64.3** | **60.2** |

### C.3 Qualitative Results

In this section, we present an additional comprehensive qualitative analysis of MPO.

**Additional Analysis in MPO**   Figure 14 illustrates the optimization process in the molecular domain, and Table 8 shows examples of textual conditions for the modality-specific generator and the resulting image prompts.

**Examples of Optimized Multimodal Prompts**   The optimized multimodal prompts from MPO are presented for the image (Table 9), video (Table 10), and molecular domains (Tables 11 and 12).

**Comparison with Text-only Optimization Baseline**   We compare the optimized prompts and responses of MPO with those of a leading text-only optimization baseline (SEE) to investigate the underlying reasons for the effectiveness of using multimodal prompts. As shown in Tables 13 and 15, text-only methods must encode all necessary visual cues solely through language, often resulting in verbose descriptions and overly generic instructions (e.g., "Pay attention to specific visual markers such as bill shape, color, and the feather crest on the Crested Auklet. . . "). This reliance on vague textual features can lead to coarse reasoning and eventual misclassification by the model (e.g., wrongly identifying the bird as a "Rhinoceros Auklet" based on an imprecise assessment of "robust body" and "horn-like projection"). In contrast, as shown in Tables 14 and 16, MPO establishes a synergistic interaction between modalities. It augments a concise textual instruction with a direct visual reference, eliminating linguistic ambiguity and instructing the model to ground its reasoning in concrete visual evidence (e.g., "compare the size and shape of the bill with those in the reference image"). This multimodal grounding shifts the model's focus, leading to more specific and visually-supported reasoning (e.g., "The bill is relatively short and thick, with a slight curve at the tip. This matches the description of the 'crested auklet' in the reference image..."). In addition to this, importantly, on examples from the Auklet subtask where SEE's optimized prompt fails, MPO's multimodal approach successfully fixes 66.6% of the misclassifications, demonstrating that multimodal cues can effectively resolve errors that are inherently difficult for text-only optimization.

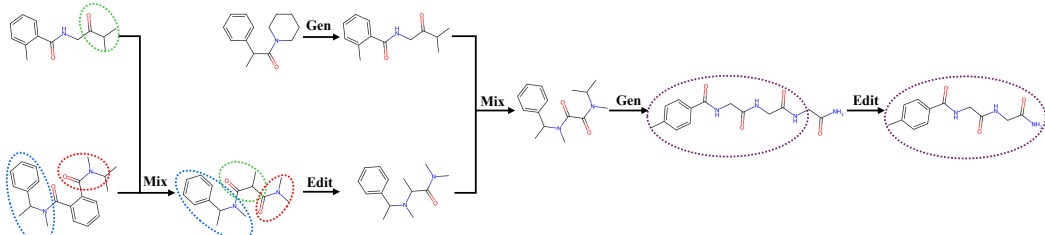

Figure 14: The optimization process for the best multimodal prompt on the BBBP task. Inherited substructures from the parent molecule are marked with the same colored circles.

## D   Use of Large Language Models (LLMs)

We use large language models merely as a writing assistant. Its role is confined to improving grammar and paraphrasing sentences for clarity, and all the core ideas regarding problem definition, MPO framework, experimental design, and interpretation of results are entirely our own.

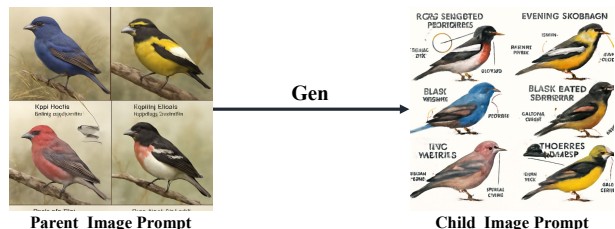

**Parent Image Prompt**        **Child Image Prompt**

$c_{\text{gen}}$: **Textual Condition for Generation Operator**

Create a detailed reference image showcasing various species of grosbeaks, ensuring each bird is clearly labeled with its common name and distinctive features. Highlight key visual characteristics such as:

- Beak shape (e.g., wide, pointed)
- Color patterns (e.g., primary colors, unique markings)
- Body size (e.g., small, medium, large)

Use arrows or circles to emphasize these features, especially where they differ from other species. The composition should be organized in a grid format for easy comparison, with a neutral background to avoid distractions from the birds themselves.

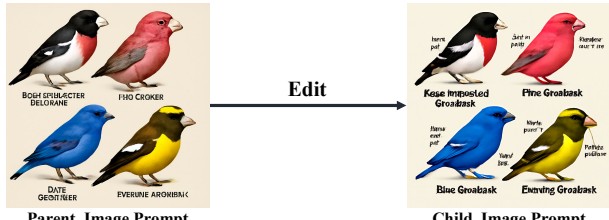

**Parent Image Prompt**        **Child Image Prompt**

$c_{\text{edit}}$: **Textual Condition for Edit Operator**

Enhance the reference image of grosbeaks by implementing the following modifications to improve clarity and usability for classification tasks:

1. **Increase Resolution and Clarity**: Use a higher-resolution image to ensure that all details of the birds are crisp and easily identifiable.
2. **Labeling**: Clearly label each grosbeak species with bold, legible text. Ensure that the labels are positioned close to the respective birds and distinguishable from the background.
3. **Consistent Postures**: Arrange the birds in similar postures and angles to facilitate direct visual comparisons. Consider a uniform side view to best showcase the beak shapes and body sizes.
4. **Highlight Distinctive Features**: Add visual cues such as arrows or circles that point to unique characteristics (e.g., beak shape, wing colors, and markings). Include brief descriptions of these features near the labels.
5. **Simplify Background**: Remove any distracting elements from the background, opting for a neutral color that allows the birds to stand out more prominently.
6. **Maintain Proportions**: Ensure that the proportions of the birds remain accurate and consistent with their actual sizes to aid in the visual comparison.

These changes aim to create a more effective reference image that enhances the classification of grosbeaks by making distinguishing features more accessible.

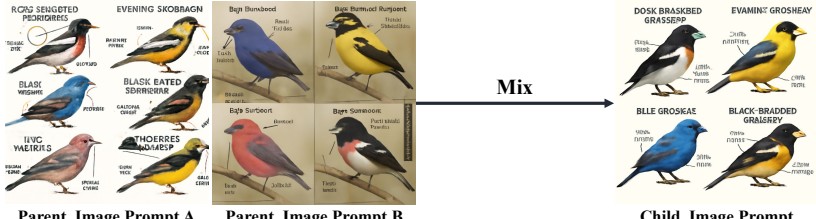

**Parent Image Prompt A**     **Parent Image Prompt B**        **Child Image Prompt**

$c_{\text{mix}}$: **Textual Condition for Mix Operator**

Create a new reference image that combines the best visual elements from both provided grosbeak images. Retain the clear anatomical labeling from Reference Image B, ensuring each grosbeak species is distinctly identified. Highlight key features such as color patterns and beak shapes using arrows and concise annotations.

From Reference Image A, incorporate the variety of grosbeak species but arrange them in a less cluttered format, allowing for a clearer comparison of unique characteristics. Focus on using high-resolution images that showcase the birds in similar poses and angles. Ensure that the primary color, distinctive markings, and beak shapes are clearly visible and easily comparable to aid in accurate classification. Discard any elements that create visual confusion or do not add value to the identification process.

Table 8: Operation examples for the image prompt update, including parent image prompts, resulting child image prompts, and the textual condition $c$ to the modality-specific generator, i.e., GPT-Image.

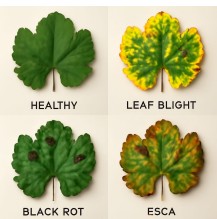

**PlantVillage**: Plant Leaf Images

Analyze the provided grape leaf image and classify it into one of the following categories: ['Healthy', 'Leaf Blight', 'Black Rot', 'Esca']. Use the hybrid reference image for guidance, focusing on the following critical visual features:

1. **Healthy**: Look for a vibrant, uniform green color and a smooth texture without blemishes.
2. **Leaf Blight**: Identify distinct yellowing edges along with well-defined small dark spots that are clearly visible.
3. **Black Rot**: Check for sharply defined, dark, sunken lesions that are prominent on the leaf surface, often accompanied by slight shriveling.
4. **Esca**: Look for distinct irregular brown patches, significant necrosis, and curling of the leaf edges.

In cases where symptoms overlap, prioritize the most severe characteristics. For example, if both dark spots and sunken lesions are present, classify based on the prominence of the lesions. Ensure that you assess each feature carefully, referencing the hybrid image to visualize these distinctions accurately.

**CUB**: Bird Images

Classify the bird in the target image by comparing it with the hybrid reference image of grosbeaks. Follow these refined steps for accurate classification.

1. **Identify the Grosbeak Group**: Refer to the hybrid image that displays the Rose Breasted Grosbeak, Pine Grosbeak, Blue Grosbeak, and Evening Grosbeak. Familiarize yourself with the specific traits of each species, including color patterns and markings.
2. **Analyze Visual Features**: Focus on these critical features of the target bird:
   – **Dominant Color and Markings**: Note the primary color and any distinctive patterns, such as throat colors or wing designs.
   – **Beak Characteristics**: Compare the shape and size of the beak with those in the reference image, as these can vary significantly among species.
   – **Body Size Comparison**: Assess the body size of the target bird relative to the reference birds, ensuring accurate size comparisons.

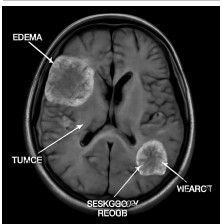
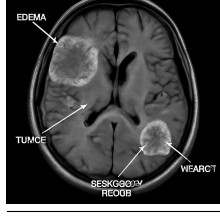

3. **Feature Prioritization**:
   – Prioritize color patterns first, as they are often the most telling feature.
   – If colors are similar, evaluate beak shape and size next.
   – Finally, consider body size. If the target bird does not closely match any reference species, provide the name of the closest match or indicate 'unknown', based on the following criteria:
      – Closeness is determined by the degree of similarity across all analyzed features, with color being the primary factor, followed by beak shape and size.

After analyzing these features, provide the name of the bird species that most closely matches the visual characteristics observed in the target image, supported by specific observations from the hybrid reference image.

**SLAKE**: Radiological Images

Given the MRI scan image, identify and list all visible abnormalities present in the brain. Your response should include specific conditions, such as edema and tumors, along with their locations (e.g., "Right Lobe" for edema). Refer to the labeled markers in the image to assist with your analysis. Aim to provide a comprehensive answer that covers all relevant conditions without omitting any visible feature. Ensure your response is clear and concise.

**DrivingVQA**: Driving Images

Examine the provided image of a road scenario and determine the most appropriate action based on specific visual cues. Pay close attention to the following details:

1. **Lane Markings**: Identify the lane markings; solid lines indicate a no-overtaking zone, while dashed lines indicate a safe area for overtaking. Clearly explain how these markings influence your decision.
2. **Position of Vehicles**: Assess the positions and distances of the vehicles. Determine if there is enough space and time to safely execute an overtaking maneuver based on their speeds and proximity.
3. **Traffic Signs**: Observe all visible traffic signs, particularly their meanings. For example, a triangular sign may indicate a hazard ahead, while a circular sign specifies speed limits. Explain how each sign influences your decision.

Based on your observations, decide whether you would (A) continue the overtaking maneuver or (B) move to the right. Justify your choice with specific details from the image, ensuring clarity in your reasoning. Conclude your response with "The answer is [answer]." Use the image as a reference to support your analysis of lane markings, vehicle positions, and relevant traffic signs in this driving scenario.

**RSVQA**: Remote Sensing Images

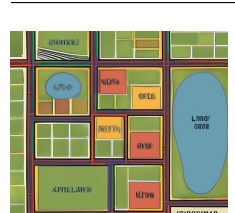

Analyze the provided neighborhood map and respond to the following questions with accurate counts and concise answers:

1. Count the total number of small roads (less than 5 feet wide), medium roads (5-10 feet wide), and large roads (greater than 10 feet wide). Indicate which category has the highest count. Use the legend provided to classify each road accurately.
2. Identify if there is a commercial building (a structure used for business purposes, such as shops or offices) located to the left of any farmland area. Ensure you consider the top-down perspective of the map when determining placement.
3. For any presence or absence questions, provide a direct "Yes" or "No" response.

Refer to the maps's colors and labels, ensuring you utilize the legend for accurate identification of each category. Pay special attention to spatial relationships as defined in the map to avoid misinterpretations.

Table 9: Qualitative examples of the optimized multimodal (image and text) prompts.

**Drive&Act**: Driver's Action Videos

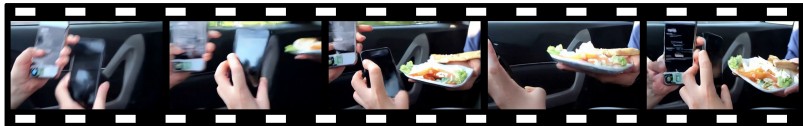

Classify the primary action being performed in the video, focusing specifically on interactions with objects or devices. If multiple actions are present, prioritize the action that is most visually prominent or contextually relevant. For example, if a person is both eating and using a phone, classify the action of using the phone. Use the definitions provided for each action to guide your decision. Consider visual cues such as hand movements, object handling, and the overall context of the scene to help determine the primary action. If two actions appear equally relevant, choose the one that is visually dominant or crucial to understanding the situation.

**VANEBench**: Abnormal Videos

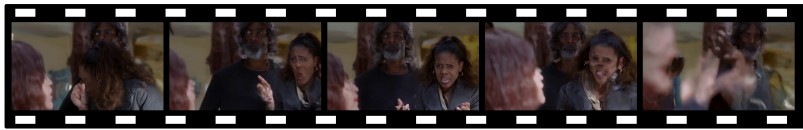

Analyze the provided video to identify and describe specific actions or behaviors depicted. Focus particularly on actions that diverge from common social norms or expectations. For instance, typical actions might include greeting someone or making eye contact, while atypical actions could involve unexpected emotional reactions, erratic movements, or interactions that seem out of place. Consider the following examples:
- A) A person suddenly laughing in a serious situation.
- B) Someone avoiding eye contact in a social setting.

Please select the most striking anomaly from the provided options and present your answer in the format: "The answer is [answer]."

Table 10: Qualitative examples of the optimized multimodal (video and text) prompts.

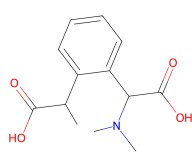

**Absorption**: Drug Absorption to Human

You are a drug discovery assistant tasked with predicting the human intestinal absorption (HIA) of a newly designed hybrid molecule. Your analysis should focus on the following physicochemical properties, taking into account both the strengths and limitations of previous reference molecules:

1. **Molecular Weight (MW):** Calculate the molecular weight of the hybrid molecule. A molecular weight below 500 Da is generally favorable for absorption.
2. **Lipophilicity (LogP):** Estimate the LogP value of the hybrid molecule. Aim for a value between -2 and 5, ensuring that it balances contributions from both polar and non-polar functional groups.
3. **Polarity and Solubility:** Analyze the overall polarity of the molecule. While polar functional groups can enhance solubility, ensure that their presence does not excessively hinder absorption through lipid-rich environments.

4. **Functional Groups:** Identify and describe key functional groups present in the hybrid molecule. Focus on ionizable groups that can enhance solubility while ensuring that non-polar groups are balanced to facilitate membrane permeability. Discuss how these groups interact to affect absorption.

5. **Stereochemistry:** Note any chiral centers present in the molecule. Different enantiomers may exhibit varying absorption profiles, so explain how stereochemistry could influence absorption.

At the end of your analysis, provide a conclusion formatted as either 'Final answer: Absorbed' or 'Final answer: Not absorbed.' Ensure that your evaluation is comprehensive and considers the combined properties derived from the reference molecules to accurately predict the absorption potential of the hybrid molecule.

Utilization of the Reference Molecule: This hybrid molecule is designed to improve predictions for human intestinal absorption (HIA) by integrating key features that enhance solubility and membrane permeability. The presence of two carboxylic acid groups increases the likelihood of ionization, which can improve solubility in the gastrointestinal tract, while the tertiary amine enhances interaction with transporters, facilitating absorption into the bloodstream.

The molecular weight is kept below 500 Da, aligning with favorable absorption criteria, and the LogP is balanced to ensure optimal lipophilicity. This design allows for a comprehensive analysis of the hybrid molecule's physicochemical properties, which can be used to inform predictive models for HIA. By leveraging the strengths of both reference molecules, the hybrid is expected to yield more accurate predictions regarding absorption potential, thereby aiding in drug discovery efforts. The combination of polar and non-polar functional groups ensures that the molecule can effectively navigate the lipid-rich environments of the intestinal membrane while maintaining sufficient solubility for absorption.

Table 11: Qualitative examples of the optimized multimodal (molecule and text) prompts.

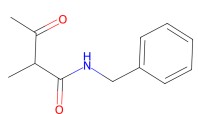

**BBBP**: Penetration to Blood-Brain Barrier

You are a drug discovery assistant responsible for predicting the blood-brain barrier (BBB) penetration capability of a new hybrid molecule based on its physicochemical properties. Follow these detailed instructions to conduct your analysis effectively:

### Key Considerations for BBB Penetration:

1. **Lipophilicity (LogP)**: Estimate the lipophilicity of the hybrid molecule. Aim for a LogP between 1 and 5, which is optimal for BBB crossing. Use computational tools like ALOGPS or ChemDraw to report the estimated LogP value.

2. **Molecular Weight**: Check the molecular weight of the hybrid molecule. It should be below 450 Da. Provide the exact molecular weight in your analysis.

3. **Hydrogen Bonding**: Assess the number of hydrogen bond donors (HBDs) and acceptors (HBAs). Aim for 1-2 HBDs and 3-5 HBAs to enhance the likelihood of BBB penetration. If the counts exceed these ranges, note how this may affect permeability.

4. **Ionization State**: Evaluate if the hybrid molecule is neutral or charged at physiological pH (~7.4). Clearly state the estimated ionization state and include pKa values for relevant groups.

5. **Presence of Polar Groups**: Identify polar functional groups and assess their overall impact on BBB permeability. Ensure a balanced presence to avoid excessive hydrophilicity.

### Analysis Instructions:
   - Analyze the provided hybrid molecular structure and evaluate how these properties collectively influence its ability to cross the BBB. Provide specific quantitative measures where applicable.
   - Compare the hybrid molecule's characteristics with those of a well-characterized reference molecule known to cross the BBB, noting key differences in properties that may influence permeability.
   - Summarize your findings clearly, stating the implications of the physicochemical properties on BBB crossing capability.

### Final Answer Format:
Conclude your analysis with a clear statement formatted as either: 'Final answer: Can cross BBB' or 'Final answer: Cannot cross BBB.'

Ensure that your analysis is thorough and based on the specific physicochemical properties outlined above to enhance the accuracy of your predictions.

Utilization of the Reference Molecule: This modified molecule enhances predictions for blood-brain barrier (BBB) penetration by addressing several key physicochemical properties.

1. **Lipophilicity (LogP)**: The addition of a methyl group (C) at the terminal position increases the hydrophobic character of the molecule, which can improve its LogP value, making it more favorable for BBB crossing. This change is aimed at achieving a LogP within the optimal range of 1-5.

2. **Molecular Weight**: The modified molecule maintains a molecular weight below 450 Da, ensuring compliance with a critical criterion for BBB penetration. This is essential as larger molecules often struggle to cross the barrier.

3. **Hydrogen Bonding**: The modification retains a balanced number of hydrogen bond donors (HBDs) and acceptors (HBAs). By keeping the HBDs to a minimum (1-2) and ensuring HBAs are within the optimal range (3-5), the likelihood of effective BBB penetration is increased.

4. **Ionization State**: The structural modifications aim to maintain the molecule in a neutral or partially ionized state at physiological pH, which is critical for enhancing lipophilicity and reducing the likelihood of charge-related hindrances to BBB penetration.

5. **Presence of Polar Groups**: The molecule has been adjusted to balance polar functional groups, reducing excessive hydrophilicity while retaining necessary polar characteristics for biological activity.

By focusing on these modifications, the molecule is better positioned for predictive modeling of BBB penetration capabilities, as it aligns with established physicochemical parameters known to influence permeability. This can inform computational predictions and improve the accuracy of models assessing BBB crossing potential.

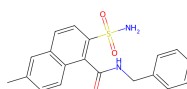

**CYP Inhibition**: Inhibitory Effect to Cytochrome P450 (CYP) Enzymes

You are a drug discovery assistant tasked with predicting the CYP2C19 inhibition potential of a target molecule. Your analysis should focus on identifying and evaluating specific structural features that correlate with CYP2C19 inhibition. Address the following key characteristics:

1. **Aromatic Rings**: Identify the number and type of aromatic rings present in the target molecule. Multiple aromatic rings are important for $\pi$-$\pi$ stacking interactions with the CYP2C19 enzyme, enhancing inhibition potential.

2. **Functional Groups**: Assess for the presence of functional groups known to enhance binding, such as:
   - **Sulfonamide groups** (–S(=O)$_2$–N–) and **amide groups** (–C(=O)N–), which facilitate hydrogen bonding and electrostatic interactions.
   - Highlight any other functional groups that may support or hinder binding.

3. **Basic Nitrogen Atoms or Heterocycles**: Determine if the molecule includes basic nitrogen atoms or heterocycles that could enhance binding affinity through electrostatic interactions.

4. **Comparison with Known Inhibitors**: Compare the structural features of the target molecule with those of known CYP2C19 inhibitors like Omeprazole or Voriconazole. Pay close attention to similarities and differences in aromaticity, functional groups, and other relevant characteristics.

Conclude your analysis with a clear statement regarding the inhibition status of the target molecule, formatted as follows: 'Final answer: Inhibits CYP2C19' or 'Final answer: Does not inhibit CYP2C19.' Ensure your comparisons and conclusions are supported by your structural analysis.

Utilization of the Reference Molecule: This generated molecule, which contains multiple aromatic rings, a sulfonamide group, and an amide group, can significantly improve predictions for CYP2C19 inhibition potential. The presence of two aromatic rings enhances $\pi$-$\pi$ stacking interactions with the CYP2C19 enzyme, which is crucial for binding affinity. The sulfonamide group (–S(=O)$_2$–N–) is known to facilitate hydrogen bonding, while the amide group (–C(=O)N–) can participate in additional hydrogen bonding interactions, both of which are important for stabilizing the enzyme-inhibitor complex.

Furthermore, the molecule incorporates basic nitrogen atoms in the amide and sulfonamide groups, which can engage in electrostatic interactions with the enzyme, further enhancing binding affinity. This structural design aligns with the characteristics observed in known CYP2C19 inhibitors like Omeprazole and Voriconazole, which also feature multiple aromatic systems and functional groups that promote hydrogen bonding.

By comparing the generated molecule's structural features with those of established inhibitors, we can derive insights into the molecular descriptors that correlate with inhibition potential. This molecule serves as a reference point for evaluating new compounds in terms of their predicted CYP2C19 inhibition, helping to refine predictive models and improve the accuracy of virtual screening processes. The combination of aromaticity, functional groups, and basic nitrogen atoms in this molecule provides a robust framework for understanding and predicting CYP2C19 inhibition.

Table 12: Qualitative examples of the optimized multimodal (molecule and text) prompts.

| Task Example: CUB(Grosbeak) Classification | |
|---|---|
| **Optimized Prompt (SEE)** *(Score: 96.7)* | Identify the species of the bird in the given image using the options provided. Please respond with the bird's name exactly as it is listed (e.g., 'rose breasted grosbeak'). 

 Make sure to write the name precisely as it appears in the choices, without any extra punctuation or modifications. Pay attention to distinctive features such as color patterns, markings, and size to accurately identify the bird. 

 If you find that the bird in the image does not match any of the listed options, please state 'not identifiable' rather than guessing. Ensure that your answer is selected only from the choices provided and refrain from inferring details about the species. 

 After determining the bird, verify that your response is among the listed options before you submit it. |
| **Image Query** 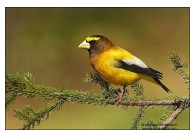 | **Ground Truth**: **Evening Grosbeak** 
 **Key Features**: 

 • White wing patches 
 • Yellow body color |
| **Response** | **rose breasted grosbeak** |

Table 13: Visualization of the SEE result on the CUB (Grosbeak) dataset.

| Task Example: CUB (Grosbeak) Classification | |
|---|---|
| **Optimized Prompt (MPO)** *(Score: 100.0)* 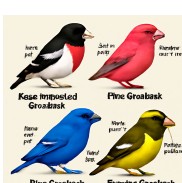 | Classify the bird in the target image by comparing it with the hybrid reference image of grosbeaks. Follow these refined steps for accurate classification. 

 1. **Identify the Grosbeak Group**: Refer to the hybrid image that displays the Rose Breasted Grosbeak, Pine Grosbeak, Blue Grosbeak, and Evening Grosbeak. Familiarize yourself with the specific traits of each species, including color patterns and markings. 

 2. **Analyze Visual Features**: Focus on these critical features of the target bird: 
   – **Dominant Color and Markings**: Note the primary color and any distinctive patterns, such as throat colors or wing designs. 
   – **Beak Characteristics**: Compare the shape and size of the beak with those in the reference image, as these can vary significantly among species. 
   – **Body Size Comparison**: Assess the body size of the target bird relative to the reference birds, ensuring accurate size comparisons. 

 3. **Feature Prioritization**: 
   – Prioritize color patterns first, as they are often the most telling feature. 
   – If colors are similar, evaluate beak shape and size next. 
   – Finally, consider body size. If the target bird does not closely match any reference species, provide the name of the closest match or indicate 'unknown', based on the following criteria: 
     – Closeness is determined by the degree of similarity across all analyzed features, with color being the primary factor, followed by beak shape and size. 

 After analyzing these features, provide the name of the bird species that most closely matches the visual characteristics observed in the target image, supported by specific observations from the hybrid reference image. |
| **Image Query** 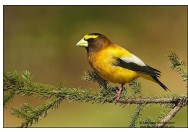 | **Ground Truth**: **Evening Grosbeak** 
 **Key Features**: 

 • White wing patches 
 • Yellow body color |
| **Response** | The bird in the target image has a predominantly yellow body with black wings and a white patch on its chest. Its beak is short and conical, suitable for seed-eating. **Comparing these features with the hybrid reference image of grosbeaks**, the bird most closely resembles the **Evening Grosbeak**. The yellow coloration, black wings, and white chest patch are characteristic of this species. Therefore, the bird in the target image is classified as an **Evening Grosbeak**. |

Table 14: Visualization of the MPO result on the CUB (Grosbeak) dataset.

| | **Task Example: CUB(Auklet) Classification** |
|---|---|
| **Optimized Prompt (SEE)** *(Score: 50.0)* | Classify the content of the target image based on the provided choices of bird species: ['parakeet auklet', 'rhinoceros auklet', 'crested auklet', 'least auklet']. 

 **Comprehensive Guidelines for Classification:** 

 **1. Physical Characteristics**: 

 **- Parakeet Auklet**: Look for a small bird (approx. 23 cm), with a striking blue bill and a notable yellow eyebrow stripe. Its compact body and vibrant colors are key identifiers. 
 **- Rhinoceros Auklet**: This medium-sized bird (about 35 cm) has a stout body and a distinctive horn-like projection on its bill during breeding season. It typically weighs around 700 grams, so note its robust build. 
 **- Crested Auklet**: Identify this species by its long, curved crest of feathers and dark plumage, roughly 30 cm in size. Weighing about 300 grams, it is often found in colonies, so observe any social behaviors. 
 **- Least Auklet**: The smallest of the auklets (around 20 cm), it features short wings and a stubby bill, weighing about 130 grams. Its petite size and subtle coloration are distinguishing traits. 

 **2. Habitat and Behavior**: Focus on typical habitats such as coastal regions and rocky cliffs. Note behaviors like diving and feeding. For instance, the **Rhinoceros Auklet** may exhibit aggressive diving, while the **Crested Auklet** shows unique social interactions. 

 **3. Visual Cues**: Pay attention to specific visual markers such as bill shape, color, and the feather crest on the **Crested Auklet**. Consistently assess the color of the feet and overall body size. 

 **4. Common Confusions**: Be aware that the **Rhinoceros Auklet** appears more robust than the slender **Least Auklet**. Additionally, plumage colors may vary under different lighting conditions. 

 **5. Cross-Verification Process**: Use a methodical approach to verify classification. For instance, check if dark plumage corresponds with the bill shape and size of the suspected species, and look for unique features. 

 **6. Key Features to Focus On**: Highlight aspects such as beak shape, feather patterns, and eye color during evaluation. 

 **7. Narrow Down Options**: If the image shows features that could apply to multiple species, focus on the most probable candidates based on observed visual cues and behaviors. 

 **8. Confidence Level**: Indicate your confidence in the classification as "certain," "likely," "uncertain," or "doubtful." Justify your confidence by detailing which specific features influenced your assessment. 

 **9. Step-by-Step Checklist for Classification**: Use this systematic checklist: Observe the size and body structure; Note the bill's color and shape; Examine plumage patterns and unique features (e.g., crests); Consider the observed habitat and behaviors; Cross-reference traits with the characteristics of all species. 

 **10. Hypothetical Example Scenarios**: Provide examples of images and a breakdown of classifications according to the guidelines, illustrating effective application of features and observations. 

 Please provide the identified species name and your confidence level in the classification. |
| **Image Query** | 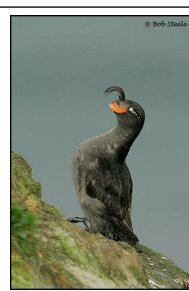    **Ground Truth**: **Crested Auklet** 
 **Key Features**: 

 • Dark plumage 
 • Curved crest of feathers 
 • Orange beak |
| **Response** | Based on the provided image, let's analyze the bird step-by-step using the comprehensive guidelines: 

 **1. Physical Characteristics**: The bird has a robust body, which aligns with the Rhinoceros Auklet being described as medium-sized (about 35 cm) and having a stout body. The bill is orange with a black tip, which is characteristic of the Rhinoceros Auklet. There is a noticeable horn-like projection on the bill, which is a key feature of the Rhinoceros Auklet during the breeding season. 

 **2. Habitat and Behavior**: The bird is perched on a rocky cliff, which is consistent with the Rhinoceros Auklet's habitat preference. The posture and behavior do not provide additional specific clues but do not contradict the identification either. 

 **3. Visual Cues**: The bill shape and color match the description of the Rhinoceros Auklet. The overall body structure and size are consistent with the Rhinoceros Auklet. 

 **4. Common Confusions**: The Least Auklet is much smaller and has a shorter bill, which does not match the image. The Crested Auklet has a long, curved crest, which is not present here. The Parakeet Auklet is smaller and has a distinct yellow eyebrow stripe, which is not seen here. 

 **5. Cross-Verification Process**: The combination of the robust body, orange bill with a black tip, and the horn-like projection strongly supports the identification of the Rhinoceros Auklet. 

 **6. Key Features to Focus On**: The bill shape and color are critical features for this identification. 

 **7. Narrow Down Options**: The image clearly matches the Rhinoceros Auklet based on the physical characteristics and visual cues. 

 **8. Confidence Level**: Given the clear match with the Rhinoceros Auklet's physical characteristics and the absence of any conflicting features, I am confident in the classification. 

 **Conclusion:** 
 The identified species is the **Rhinoceros Auklet**, and my confidence level in this classification is **certain**. |

Table 15: Visualization of the SEE result on the CUB (Auklet) dataset.

| Task Example: CUB(Auklet) Classification |
|---|

**Optimized Prompt (MPO)** *(Score: 80.3)*

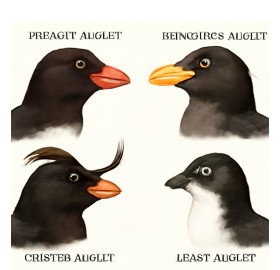

Analyze the provided hybrid image of auklet species and classify the bird in the target image into one of the following categories: ['parakeet auklet', 'rhinoceros auklet', 'crested auklet', 'least auklet']. Focus on the following critical visual features:

1. **Bill Shape**: Prioritize the size and shape of the bill, comparing it directly with the examples in the hybrid image. Be specific about curvature and thickness.

2. **Plumage Color**: Observe the overall body color and any variations, noting how these compare to the reference images.

3. **Body Patterns**: Identify unique markings or patterns on the body, ensuring to highlight any significant differences that may aid in classification.

Use the hybrid image as a guide for your comparison and classification. Clearly mention any distinguishing traits that support your decision, and ensure to prioritize bill shape in cases of ambiguity.

**Image Query**

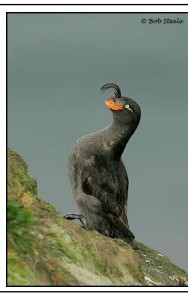

**Ground Truth**: **Crested Auklet**
**Key Features**:

- Dark plumage
- Curved crest of feathers
- Orange beak

**Response**

The bird in the target image can be classified as a "crested auklet" based on the following observations:

1. **Bill Shape**: The bill is relatively short and thick, with a slight curve at the tip. This matches the description of the "crested auklet" in the reference image, which has a similar bill shape but with a more pronounced curve. The "rhinoceros auklet" also has a thick bill but with a more pronounced curve, while the "parakeet auklet" and "least auklet" have thinner bills.

2. **Plumage Color**: The overall body color is dark, almost black, with some lighter areas around the neck and underparts. **This is consistent with the "crested auklet" in the reference image.** The "rhinoceros auklet" has a similar dark plumage, but the "parakeet auklet" and "least auklet" have lighter plumage.

3. **Body Patterns**: The most distinctive feature of the "crested auklet" is the presence of a crest on its head, which is clearly visible in the target image. The "parakeet auklet" and "least auklet" do not have crests, and the "rhinoceros auklet" has a shorter crest compared to the "crested auklet".

In conclusion, the combination of the short and thick bill, dark plumage, and the presence of a crest on the head strongly supports the classification of the bird in the target image as a **"crested auklet"**.

Table 16: Visualization of the MPO result on the CUB (Auklet) dataset.

