# OpenReview forum: "Multimodal Prompt Optimization: Why Not Leverage Multiple Modalities for MLLMs"
_ICLR.cc/2026/Conference — ICLR 2026 Poster_

### Official Review · Reviewer_DhB1 · 2025-10-27

**Soundness:** 3
**Presentation:** 3
**Contribution:** 3
**Rating:** 6
**Confidence:** 3

**Summary:**

The paper proposes the Multimodal Prompt Optimizer (MPO), which jointly refines textual and non-textual prompts through alignment-preserving updates and employs a Bayesian UCB strategy for efficient candidate selection. Experiments across images, videos, and molecules show that MPO outperforms text-only baselines, highlighting multimodal prompt optimization as a key step toward advancing MLLMs.

**Strengths:**

- The paper expands the prompt optimization space beyond text to incorporate multiple modalities, which represents an interesting and promising research direction.
- MPO shows strong performance across diverse modalities (images, videos, molecules), demonstrating its effectiveness.
- The paper is clearly written and easy to follow.

**Weaknesses:**

Although the paper claims that MPO is not an Instance-Specific Prompting and Optimization method but rather aims to discover a single, reusable prompt that enhances performance across an entire task, the evaluation still relies on fine-grained dataset partitioning and reducing the number of classes to control task difficulty. This design choice appears to weaken the claimed generalization ability of the proposed method, suggesting that its effectiveness in more complex or diverse scenarios remains to be validated.

**Questions:**

- Is there a clear order in which the three Exploration Operators designed in this paper are called, and how are they combined? Figure 6 shows that there doesn't appear to be a clear order in which operators are called, and the combinations are arbitrary. The original paper mentions that these operators systematically expand, refine, and recombine non-textual prompts. Could you explain this systematic approach in more detail?
- In the Appendix (lines 665–670), the authors mention that for certain datasets, they further selected groups containing three or four distinct species to maintain a balanced level of difficulty. However, this strategy effectively reduces the overall task complexity, as the generated multimodal prompts only need to distinguish among a small number of categories. It is recommended that the authors further discuss whether the proposed method can still maintain good performance and stability when the number of categories increases significantly.
- The generation of non-textual multimodal prompts in this paper appears to rely heavily on high-performance image generation and editing models. It is suggested that the authors clarify the extent to which their method depends on the specific capabilities of these models. If alternative image generation or editing models were used, would the proposed approach still maintain its effectiveness and robustness? If the authors can provide convincing explanations or additional evidence regarding these questions, I would be willing to consider increasing the score.

---

> ### Author Response · Authors · 2025-11-21
> **Response to Reviewer DhB1**
>
> We sincerely appreciate your thoughtful and encouraging feedback. We are grateful that you recognized the value of multimodal prompt optimization as a promising direction, the strong performance of MPO across images, videos, and molecules, and the clarity of presentation. We have carefully considered and faithfully responded to all your comments below.
>
> ---
>
> > W1. While MPO is claimed not to be an instance-specific prompting and optimization method, the evaluation relies on fine-grained datasets partitioning and reducing the number of classes to control task difficulty, which weakens its claimed generalization ability.
>
> We appreciate your constructive comment. We would like to first note that MPO is not an instance-specific method; rather, it is designed to discover a single, reusable multimodal prompt that is optimized at the task level, similar to other prompt optimization methods, and moreover, our experiment is designed to measure the performance at the task level. Furthermore, while some datasets in our evaluation suite include category structures, several of our benchmarks, such as SLAKE and RSVQA (open-ended QA), as well as DrivingVQA (multiple-choice QA), are not class-dependent tasks, which further require reasoning over semantically diverse textual and non-textual query pairs (see Appendix A.1). In addition to this, one of our datasets (namely, Drive&Act) provides a challenging setting with 34 distinct classes. Lastly, across all of these settings, MPO outperforms text-only baselines (Table 1), and qualitative analyses in Tables 9 and 10 further confirm that the optimized multimodal prompts capture task-relevant semantics (rather than instance-specific or class-specific patterns). In other words, given them, we believe that MPO genuinely learns generalizable, task-level multimodal prompts.
>
> ---
>
> > Q1. Is there a clear order in which the three Exploration Operators designed in this paper are called, and how are they combined?
>
> This is a great question. Our three exploration operators are all applied uniformly at each iteration, and their outputs are then passed to the Bayesian UCB-based selection stage, which determines which candidates survive to the next round. In this context, the combination of operators naturally emerges over iterations: at each step, diverse modifications are produced by different operators that coexist, and only the ones empirically shown to improve performance are propagated forward. Notably, this design allows the operators to comprehensively explore the multimodal prompt space without requiring a hand-crafted order, which would otherwise depend on prior knowledge about datasets or modalities (which might be unavailable and can vary across different settings). Also, relying on a fixed order would bias the search toward a particular type of modification or a specific dataset, rather than adapting to the characteristics of the task.
>
> ---
>
> > Q2. The authors could further discuss whether the proposed method can still maintain good performance and stability when the number of categories increases.
>
> We thank you for raising this important point, and we have faithfully addressed it in our response to W1, discussing the performance of MPO on complex tasks with a large-class setting; thus, we kindly ask you to refer to our response in W1.
>
> ---
>
> > Q3. Effectiveness and robustness of MPO across diverse modality-specific generators.
>
> Thank you for pointing it out. We agree with the importance of evaluating MPO across diverse modality-specific generators, and in fact, we already conducted experiments to assess this. Specifically, as shown in Table 2 (Bottom Right), MPO maintains strong performance even when paired with lightweight and open-source alternatives: for instance, using the small open-source SANA1.5 (1.6B) for image generation, MPO (71.8) still significantly outperforms the strong text-only baseline SEE (69.0). While a more capable generator does yield better scores (e.g., 76.6 with GPT-Image-Medium), the consistent improvement across different generators demonstrates that the performance lift is driven by its joint, alignment-preserving optimization strategy with prior-inherited selection, rather than from the raw capacity of a specific (and powerful) generator.

---

> > ### Comment · Reviewer_DhB1 · 2025-11-27
> >
> > I appreciate the author's further clarification, which resolved most of my confusion and concerns about the paper. Given that I have already given a positive review, I decided to retain my original rating.

---

> > > ### Author Response · Authors · 2025-11-28
> > >
> > > Thank you for your continued support and for acknowledging our clarification. We are glad that our response resolved most of your confusion and concerns. If there are any remaining concerns or minor points that are still unclear, please let us know. We would be more than happy to provide further explanations.

---

### Official Review · Reviewer_wMqy · 2025-10-28

**Soundness:** 3
**Presentation:** 3
**Contribution:** 3
**Rating:** 6
**Confidence:** 3

**Summary:**

This paper targets the prompt optimization of MLLMs. Unlike prior approaches that only optimize text prompts, the authors incorporate information from another modality into the prompt.
To optimize the multi-modal prompt, the authors propose MPO which consists of two key components: alignment-preserving exploration of multimodal prompt space and prompt selection with prior-inherited Bayesian UCB.

**Strengths:**

1. The idea is good, which revolutionizes the conventional prompt structure for MLLMs.
2. The paper is well-written, with clear motivation and challenges.
3. The experiments are thorough.

**Weaknesses:**

1. In the first paragraph of section 3.2, the authors mentioned that "a naive approach that independently updates textual and non-textual components risks producing misaligned prompts". Have the authors tested this naive approach (is it Random Image Prompt in Figure 4? If yes, what are the real performance values instead of performance gains?).
2. When identifying the failure set, how is your multimodal prompt initialized? As it contains both text and image, how are they selected and organized?
3. Another concern is the efficiency: extra models are included to refine the multi-modal prompt. Compared to the baseline methods which optimize textual prompt only, what is the time (or computation steps) used by MPO to complete one iteration?

**Questions:**

See weaknesses.

---

> ### Author Response · Authors · 2025-11-21
> **Response to Reviewer wMqy**
>
> We thank you for the thoughtful and insightful feedback, as well as your recognition of the strengths of our work: the idea revolutionizes the conventional prompt structure for MLLMs, its motivation and challenges are clearly presented, and the experimental evaluations are thorough. We have carefully taken your comments into account in our response below.
>
> ---
>
> > W1. Have the authors tested the naive approach that independently updates textual and non-textual components?
>
> Yes, we evaluated a naive variant (denoted as “Sequential” in Figure 4) that independently optimizes the textual and non-textual prompts (optimizing the textual prompt first, then the non-textual prompt afterward). The Sequential variant gets an absolute performance of 71.4, which is a lower performance gain than our joint MPO framework (76.4). This result indicates that such an independent optimization strategy is suboptimal, as it fails to maintain strong cross-modal alignment, whereas MPO can address this through alignment-preserving joint updates.
>
> ---
>
> > W2. When identifying the failure set, how is your multimodal prompt initialized? As it contains both text and image, how are they selected and organized?
>
> At the start of optimization (the first iteration), multimodal prompts do not yet exist; thus, we initialize the process using only the initial textual prompts (provided manually). Specifically, we run the base MLLM with these text prompts, identify all incorrect responses, and treat them as the failure set. This failure set is then fed into the cohesive backpropagation step together with the meta-prompt shown in Figure 10. Notably, it is designed to operate even when no non-textual inputs are provided. In addition to this, to construct the initial multimodal prompts, we use the generation operator (excluding edit and mix), since it does not rely on non-textual prompts (and is therefore suitable for the initial multimodal prompt generation). Then, in the subsequent iterations, the generated multimodal prompts are utilized.
>
> ---
>
> > W3. Extra models are included to refine the multi-modal prompt. Compared to the text-only baselines, what is the time (or computation steps) used by MPO to complete one iteration?
>
> Thank you for raising the efficiency concern. While MPO incurs a higher per-iteration cost than text-only baselines (i.e., text-only PE2, APO, SEE, and our MPO require 0.083, 0.071, 0.019, and 0.169 dollars of monetized API cost per iteration, respectively), per-iteration cost is, in fact, not the most meaningful metric for evaluating overall efficiency. Instead, what matters in practice is how quickly the optimizer reaches high performance under a fixed computational (or monetary) budget. To assess this, we fix the cost budget (\\$0.25 and \\$0.50) and terminate optimization once the cost is exhausted. As shown in Table D.1, MPO consistently outperforms text-only methods within the same budget, and notably, MPO with a smaller cost budget (\\$0.25) already achieves performance comparable to (or even exceeding) baselines using a higher budget (\\$0.50), demonstrating its cost-efficiency.
>
> Table D.1. Performance comparison on PlantVillage under budget constraints.
> | Method | Accuracy (%) @ \\$0.25 | Accuracy (%) @ \\$0.50 |
> |---|---|---|
> | PE2 | 64.4 | 66.7 |
> | ProTeGi | 62.4 | 63.4 |
> | SEE | 69.0 | 69.7 |
> | MPO | **70.0** | **73.5** |

---

### Official Review · Reviewer_oxEi · 2025-10-28

**Soundness:** 2
**Presentation:** 3
**Contribution:** 3
**Rating:** 6
**Confidence:** 3

**Summary:**

This paper introduces Multimodal Prompt Optimization (MPO), a new framework that extends traditional text-only prompt optimization to multimodal large language models (MLLMs) by jointly optimizing textual and non-textual components of prompts. Specifically, the method combines alignment-preserving exploration, which updates text and image prompts coherently based on failure-driven feedback, with a prior-inherited Bayesian UCB selection strategy that leverages parent–child prompt priors to improve sample efficiency. Experiments across image, video, and molecular tasks show that MPO outperforms strong text-only baselines while significantly reducing evaluation cost, suggesting that multimodal prompt optimization can better unlock the reasoning potential of MLLMs.

**Strengths:**

1.The combination of alignment-preserving exploration with prior-inherited Bayesian UCB is technically sound and elegantly balances multimodal consistency with sample efficiency.

2.The paper is clearly written and well-structured, with intuitive figures and pseudocode that align closely with the algorithmic flow, accompanied by informative ablations and visual analyses.

**Weaknesses:**

1.The paper frames “multimodal prompt optimization” as discrete text + generated visual prototypes, which risks conflation with learnable soft-prompting (e.g., MaPLe) and needs clearer terminology and positioning.

2.The notion of “parent prompt” and its construction (especially for multi-parent mix) is not formalized.

3.Including powerful image/video generators injects generator priors and extra compute, so gains may stem from tooling rather than the method itself.

4.Efficiency is measured via evaluation counts rather than real cost, omitting overheads from generation and long-context prompting.

5.Cross-modal alignment relies on a single metric (e.g., DSG).

**Questions:**

1.Which posterior quantile is used for Bayes-UCB, and how do your theoretical conditions differ from standard Bayes-UCB assumptions?

2.If you replace generated prototypes with retrieved in-distribution examples or OOD random images, how do performance and alignment change?

---

> ### Author Response · Authors · 2025-11-21
> **Response to Reviewer oxEi (1/3)**
>
> We thank you for the constructive and detailed evaluation of our work. We are encouraged that you found our methods to be technically sound and effective, and that you appreciated the clear structure of the paper, including figures, pseudocode, and experimental designs. We have carefully considered your comments and made every effort to address them.
>
> ---
>
> > W1. The paper frames “multimodal prompt optimization” as discrete text + generated visual prototypes, which risks conflation with learnable soft-prompting (e.g., MaPLe) and needs clearer terminology and positioning.
>
> Thank you for your valuable suggestion. In our work, multimodal prompts refer specifically to discrete textual prompts paired with discrete non-textual modality inputs (e.g., images, videos, molecules) that are instantiated as actual inputs to MLLMs, not continuous embeddings. This is distinct from learnable soft prompts such as MaPLe, which operate in the continuous embedding space and are optimized via gradient descent. To avoid confusion, we have added a precise definition of “multimodal prompt” in a footnote in the introduction.
>
> ---
>
> > W2. The notion of “parent prompt” and its construction (especially for multi-parent mix) is not formalized.
>
> Thank you for raising this point. In our framework, a parent prompt refers to any prompt ($p = (t, m)$) that is selected in the top-b beam at the previous iteration and thus serves as the source, from which new candidate (child) prompts ($p' = (t', m')$) are generated. For the generation and edit operations, we randomly select one parent prompt to generate a child prompt ($p \rightarrow p’$). For the mix operation, we instead sample a set of parent prompts from the top-b beam and combine them to form a mixed child prompt ($ \\{ p_1,...,p_K \\} \rightarrow p’$). We have updated the manuscript to explicitly define the notion of a parent prompt in the “Joint Optimization of Multimodal Prompt” paragraph and to clarify the parent prompt selection procedure in “Exploration Operators”.
>
> ---
>
> > W3. Including powerful image/video generators injects generator priors and extra compute, so gains may stem from tooling rather than the method itself.
>
> We appreciate this insightful comment. We also considered the possibility that our improvements might stem primarily from stronger image (or video) generators rather than the MPO method itself, and to disentangle these effects, we conducted a generator-strength analysis (Table 2, bottom right). From this, the results show that MPO’s effectiveness is not contingent on using state-of-the-art, proprietary generators. Even when instantiated with a lightweight, open-source generator (SANA 1.5, 1.6B), MPO achieves an accuracy of 71.8%, which significantly outperforms the strongest text-only baseline, SEE (69.0%). While scaling up to a more powerful generator (e.g., GPT-Image-Medium) does further improve performance to 76.6%, the consistent gains with a significantly weaker generator indicate that the improvements are not merely artifacts of generator priors. Instead, the design of MPO, particularly its multimodal exploration and prior-inherited Bayesian selection, is the primary contributor to the performance, independent of the specific generator used.

---

> ### Author Response · Authors · 2025-11-21
> **Response to Reviewer oxEi (2/3)**
>
> > W4. Efficiency is measured via evaluation counts rather than real cost, omitting overheads from generation and long-context prompting.
>
> We thank the reviewer for raising this important point about real cost. In addition to reporting evaluation costs (in Appendix C.1), we have further conducted an analysis, measuring the real cost based on the cost-constrained comparison on PlantVillage: we explicitly monetize the total budget with the API costs under fixed budgets (\\$0.25 and \\$0.50). As shown in Table C.1, MPO achieves the best accuracy within the same budget, and more notably, MPO’s performance with a small budget (\\$0.25) is comparable to or even better than text-only optimization methods with a large budget (\\$0.50). This demonstrates that MPO is a cost-efficient optimization method, achieving high performance at a smaller cost. Moreover, as highlighted in Table 2 (bottom right) of the paper, MPO remains effective with lightweight open-source generators (e.g., SANA-1.5 1.6B), whose cost is negligible relative to the performance gains they unlock. We have added the new results and highlighted efficiency.
>
> Table C.1. Performance comparison on PlantVillage under budget constraints.
> | Method | Accuracy (%) @ \\$0.25 | Accuracy (%) @ \\$0.50 |
> |---|---|---|
> | PE2 | 64.4 | 66.7 |
> | ProTeGi | 62.4 | 63.4 |
> | SEE | 69.0 | 69.7 |
> | MPO | **70.0** | **73.5** |
>
> ---
>
> > W5. Cross-modal alignment relies on a single metric (e.g., DSG).
>
> We prioritize DSG over global metrics like CLIP or BLIP, not only because DSG is a standard metric to measure cross-modal alignment (following existing works [1, 2]), but also because those global metrics are insufficiently sensitive to the attribute-level changes that are central to multimodal prompt optimization. To be more specific, DSG decomposes the textual description into atomic, dependency-aware queries and verifies each using an MLLM, enabling a precise assessment of whether the visual content faithfully reflects the intended textual details, unlike global metrics that produce high scores even when key semantic attributes are missing. We have clarified this rationale in the revised version of the paper.
>
> [1] Mañas et al., Improving Text-to-Image Consistency via Automatic Prompt Optimization. Trans. Mach. Learn. Res., 2024.
>
> [2] Kim et al., Reward-Agnostic Prompt Optimization for Text-to-Image Diffusion Models, 2025.

---

> ### Author Response · Authors · 2025-11-21
> **Response to Reviewer oxEi (3/3)**
>
> > Q1. Which posterior quantile is used for Bayes-UCB, and how do your theoretical conditions differ from standard Bayes-UCB assumptions?
>
>
> Thank you for the detailed question. As shown in Algorithm 2, our Bayes-UCB uses the upper posterior quantile at level $q\_t = 1 − \frac{1}{t(log N)^c}$, and at round $t$ we select the arm with the largest $\text{BetaQuantile}(q\_t; \alpha\_i, \beta\_i)$. Our theoretical analysis otherwise follows the standard Beta-Bernoulli Bayes-UCB [3] setting, but differs in the assumption on the prior. Instead of adopting a fixed uninformative prior, we analyze prior-inherited posteriors and assume that, on average over child arms, the parent’s posterior mean is KL-closer to the true mean than the uninformative prior mean 1/2 (Eq. (3), “average KL-closeness” assumption). While this is the only additional condition beyond standard Bayes-UCB, it brings us the huge advantage: the best-arm identification cost with prior inheritance is never worse than in the uninformative-prior case (Proposition 3.1).
>
> [3] Kaufmann et al., On Bayesian Upper Confidence Bounds for Bandit Problems, International Conference on Artificial Intelligence and Statistics, 2012.
>
> ---
>
> > Q2. If you replace generated prototypes with retrieved in-distribution examples or OOD random images, how do performance and alignment change?
>
> Thank you for this thoughtful question. In Figure 4, the In-distribution Image Query corresponds to replacing our generated prototypes with in-distribution examples, and the OOD Image Query corresponds to using random OOD images. We observe that both variants substantially reduce cross-modal alignment and decrease performance gains compared to MPO. Nevertheless, In-distribution Image Query and OOD Image Query achieve the performance of 61.0 and 59.1, still performing better than the text-only prompt optimization baselines such as APE (55.8), EvoPrompt (56.1), and OPRO (54.1). This indicates that leveraging multiple modalities and incorporating additional on-textual information remains beneficial even when the visual signals are suboptimal.

---

### Official Review · Reviewer_oPXv · 2025-10-31

**Soundness:** 2
**Presentation:** 2
**Contribution:** 3
**Rating:** 4
**Confidence:** 3

**Summary:**

This paper focuses on multimodal prompt optimization for Multimodal Large Language Models (MLLMs), addressing two critical gaps in existing text-only Automatic Prompt Optimization (APO): underutilization of MLLMs’ multimodal capabilities, and inherent challenges (cross-modal inconsistency, sparse high-quality candidates) in multimodal prompt spaces.

Key contributions:
1. Formalizes the multimodal prompt optimization problem, defining optimal prompts as text-nontext pairs \((t,m)\) that maximize MLLM performance on target tasks.
2. Proposes the **MPO framework**:
   - *Alignment-Preserving Exploration* (via cohesive backpropagation, joint multimodal update, and 3 complementary operators) ensures text-image semantic consistency;
   - *Prior-Inherited Bayesian UCB* leverages parent-child prompt performance correlation to solve cold-start, reducing evaluation budget by 42%-70%.
3. Validates on 10 datasets across 3 modalities (image/video/molecule), outperforming text-only APO (e.g., +8.6% accuracy on CUB-200-2011).

MPO fills the multimodal APO gap for MLLMs, with strong cross-model/multimodal generalization and practical efficiency.

**Strengths:**

- **Originality**: Breaks the text-only limitation of existing APO methods, first formalizing the multimodal prompt optimization problem (defining prompts as text-nontext pairs \((t,m)\)). It also creatively improves Bayesian UCB by leveraging parent-child prompt performance correlation to solve cold-start, extending bandit-based selection to multimodal scenarios innovatively.
- **Quality**: Conducts rigorous validation—covering 10 datasets across 3 modalities (image/video/molecule), cross-model tests (Qwen2.5-VL, Gemma3), and ablation studies (verifying alignment mechanisms/operators’ necessity). Results are reliable and generalize well.
- **Clarity**: Clearly presents problem formulation, MPO’s two core components (with formulas and flowcharts), and experimental design. The logical structure is straightforward, enabling easy understanding of the framework.
- **Significance**: Fills the gap of multimodal prompt optimization for MLLMs, reduces evaluation budget by 42%-70% for practicality, and provides a foundational framework for future multimodal prompt research.

**Weaknesses:**

1. **Lack of Validation on Mainstream MLLM General Benchmarks, Limiting Evidence of Universal Adaptability**
The current experiments rely solely on custom task-specific datasets (e.g., CUB-200-2011, PlantVillage) and fail to validate MPO on widely recognized MLLM multimodal benchmarks, leaving its ability to enhance MLLMs’ general capabilities unsubstantiated. Specifically:
- It omits **static multimodal foundational benchmarks** (e.g., MME, MMBench), which focus on core perceptual capabilities of MLLMs (e.g., image-text matching, attribute recognition)—there is no evidence that MPO can optimize prompt performance for these fundamental tasks.
- It excludes **long-video dynamic modality benchmarks** (e.g., VideoMME), which require handling temporal alignment between long-sequence videos and text (e.g., locating specific clips, understanding temporal logic). The paper’s existing alignment mechanism, designed for static images, remains untested for such long-video scenarios, casting doubt on its effectiveness.
- It neglects **cross-modal complex reasoning benchmarks** (e.g., ScienceQA, MathVista, Geometry3k)—benchmarks that demand MLLMs integrate multimodal information to solve logical reasoning or mathematical problems. Since the paper only validates classification/prediction tasks, there is no proof that MPO-optimized prompts can improve complex reasoning performance, leaving MPO’s adaptability to general MLLM scenarios unconfirmed.

2. **Unaddressed Implicit Deployment Costs, Missing Cost Comparison with Text-only APO**
While the paper emphasizes that the prior-inherited Bayesian UCB reduces evaluation budget by 42%–70%, it overlooks the significant computational overhead of generating multimodal candidates. Modal-specific generators (e.g., GPT-Image, video editing models) used for image/long-video clip generation incur much higher costs than text prompt generation: for instance, single-image generation takes 2–5 seconds (vs. 0.1 seconds for text generation), and diffusion-based image generators require 3–5 times more GPU memory than text models. Critically, the paper fails to compare the **total deployment cost of MPO** (including iterative multimodal generation overhead) with that of text-only APO. If the implicit costs of multimodal generation offset or even exceed the saved evaluation budget, MPO’s practical utility in real-world deployment would be severely undermined—this key cost trade-off is entirely unaddressed.

**Questions:**

Same as the section of **Weaknesses**.

---

> ### Author Response · Authors · 2025-11-21
> **Response to Reviewer oPXv (1/2)**
>
> We thank the reviewer for the careful and detailed evaluation of our work. We are glad that you acknowledge our contributions, including the formalization of multimodal prompt optimization, the proposed MPO framework, and the breadth of experimental results. We are also grateful that you view MPO as filling an important gap in multimodal prompt optimization for MLLMs. We have carefully considered your comments and faithfully responded to them.
>
> ---
>
> > W1. Lack of validation on mainstream MLLM benchmarks limits evidence of adaptability.
>
> We thank you for raising this concern. However, our benchmark suite is, in fact, considerably broader in modality coverage and task diversity compared to existing prompt optimization works. Specifically, recent prompt optimization studies (such as ProTeGi and PE2) evaluate on 4 and 6 benchmarks, all within a single modality (text-only), in contrast to our experiments that span 10 datasets across three different modalities (images, videos, and molecules). Therefore, our experimental coverage is already at least comparable to, and broader than, prior works, which we believe is sufficient for validating the effectiveness of the MPO.
>
> > W1.1. It omits static multimodal foundational benchmarks, which focus on core perceptual capabilities of MLLMs.
>
> We agree that mainstream multimodal benchmarks are valuable, and we would like to clarify that our evaluation already covers them, targeting the same core perceptual capabilities of MLLMs that such benchmarks are designed to measure. To mention a few, our image benchmarks include fine-grained recognition (CUB and PlantVillage), attribute recognition (SLAKE), and visual grounding (RSVQA), which collectively cover the visual understanding skills assessed in static multimodal benchmark suites, discussed in Appendix A.1.
>
> > W1.2. Exclusion of long-video dynamic modality benchmarks leaves MPO’s alignment mechanism untested for temporal alignment between long-sequence videos and text.
>
> Our current evaluation suite goes beyond static images and already includes settings with dynamic visual inputs. In particular, Drive&Act and VANE-Bench both consist of multi-frame clips with non-trivial temporal structure (dynamic activities, abnormal motion, temporal distortions), and we show the effectiveness of MPO in these benchmarks, demonstrating its applicability to the video sequences with temporal dependencies. Additionally, while we do not include hour-scale long-video datasets, these benchmarks differ primarily in sequence length rather than the underlying alignment challenge, and their substantially higher processing cost makes them less aligned with the scope of our work, which targets a broad, multimodality-focused evaluation rather than deep exploration of a single high-cost modality for optimization. In other words, with our current empirical results on the effectiveness of MPO to temporally structured video inputs, we view its further long-video evaluation as a promising extension rather than a requirement for validating the core mechanism of MPO.
>
> > W1.3. It neglects cross-modal complex reasoning benchmarks that demand MLLMs to integrate multimodal information to solve logical reasoning or mathematical problems.
>
> Thank you for raising this point about cross-modal complex reasoning. We would like to first note that our evaluation also includes tasks that require multimodal reasoning beyond simple perceptual matching. For instance, DrivingVQA involves causal reasoning over dynamic driving scenes, and molecular tasks (such as BBBP) require the model to integrate structural representations with domain-specific chemical knowledge and its corresponding reasoning. However, to further address the concern about mathematical problems, we have additionally conducted experiments on the Geometry3K [1] benchmark (see Table B.1), which enables testing multimodal geometric reasoning. The results show that MPO effectively optimizes multimodal prompts for these geometric problems, achieving superior performance over baselines, validating its robustness with multimodal information for complex logical tasks.
>
> Table B.1. Performance comparison on the Geometry3K benchmark.
> |  |  | Geometry3k |  |  |
> |:---:|:---:|:---:|:---:|:---:|
> |  | triangle | parallelogram | trapezoid | Avg. |
> | Human | 28.5 | 34.4 | 58.3 | 40.4 |
> | SEE | 29.4 | 36.1 | 62.5 | 42.7 |
> | MPO | **30.3** | **37.7** | **66.7** | **44.9** |
>
> [1] Lu, Pan et al., Inter-GPS: Interpretable Geometry Problem Solving with Formal Language and Symbolic Reasoning, ACL, 2021

---

> ### Author Response · Authors · 2025-11-21
> **Response to Reviewer oPXv (2/2)**
>
> > W2. Missing deployment costs of MPO and their comparisons with text-only APO.
>
> We thank the reviewer for highlighting the efficiency aspect. As you pointed out, while multimodal generation naturally introduces some overhead, we find that MPO remains the most cost-effective solution when considering the full deployment budget. To support this, we have conducted a cost-constrained evaluation on the PlantVillage dataset by fixing the total budget (i.e., monetary budget for the API costs) at fixed intervals (\\$0.25 and \\$0.50). As shown in Table B.2, MPO achieves the best performance at every budget level. Also, remarkably, MPO at a smaller budget (\\$0.25) is comparable to the performance of the state-of-the-art text-only optimization model (SEE) operating at double the cost (\\$0.50), demonstrating substantially superior cost-efficiency. Further, as shown in Table 2 (bottom right) of the paper, MPO remains effective with lightweight open-source generators (e.g., SANA-1.5 1.6B), whose cost is negligible relative to the performance gains they unlock.
>
> Table B.2. Performance comparison on PlantVillage under budget constraints.
> | Method | Accuracy (%) @ \\$0.25 | Accuracy (%) @ \\$0.50 |
> |---|---|---|
> | ProTeGi | 62.4 | 63.4 |
> | PE2 | 64.4 | 66.7 |
> | SEE | 69.0 | 69.7 |
> | MPO (Ours) | **70.0** | **73.5** |

---

### Official Review · Reviewer_z13R · 2025-10-31

**Soundness:** 3
**Presentation:** 3
**Contribution:** 3
**Rating:** 6
**Confidence:** 4

**Summary:**

This paper defines the new problem of multimodal prompt optimization and introduces MPO, a framework that jointly optimizes textual and non-textual prompts for multimodal large language models. The method combines alignment-preserving exploration across modalities with a prior-inherited Bayesian UCB strategy for efficient prompt selection. Experiments on diverse modalities (images, videos, and molecules) show consistent improvements over existing text-only prompt optimization baselines.

**Strengths:**

(1) The idea of extending prompt optimization beyond text is timely and relevant, filling a gap in the growing MLLM literature.

(2) The paper is technically clear, well-written, and supported by convincing experiments on multiple modalities and model backbones.

(3) The proposed Bayesian prior mechanism for efficient search adds a nice practical touch that improves optimization stability.

**Weaknesses:**

(1) The conceptual jump from text-only to multimodal prompt optimization is natural but not as novel as it’s presented; many parts resemble standard multimodal conditioning or input co-optimization.

(2) The analysis lacks stronger insights into why multimodal prompts help; results show improvement but don’t probe interpretability, modality interactions, or failure cases.

**Questions:**

(1) How much of the gain actually comes from the added non-textual modality rather than the optimization process itself? A controlled text-only ablation using the same search procedure would clarify this.

(2) How robust is the “alignment-preserving” exploration? When the generated visual component drifts semantically, does the optimization recover or collapse?

(3) The Bayesian UCB prior sounds appealing, but how sensitive is the performance to the prior strength? Could it bias the search toward mediocre parents if the correlation assumption breaks?

(4) Since the framework depends on GPT-based visual and molecular generators, how reproducible is this pipeline for researchers without access to those models?

---

> ### Author Response · Authors · 2025-11-21
> **Response to Reviewer z13R (1/2)**
>
> We sincerely thank you for your insightful and encouraging comments. We are grateful that you found our idea of extending prompt optimization beyond text timely and relevant, and the paper clear, empirically supported by performance gains on multiple modalities. In this response, we have carefully considered your comments and faithfully responded to them.
>
> ---
>
> > W1. The conceptual jump from text-only to multimodal prompt optimization is natural but not as novel as it’s presented; many parts resemble standard multimodal conditioning or input co-optimization.
>
> We thank you for raising this concern and would like to clarify our contributions that go beyond simply adding another modality: our novelty lies not only in formulating the task of multimodal prompt optimization itself, but also in identifying the unique technical challenges that arise from it, as follows:
>
> * First, at the task-level, unlike standard multimodal conditioning or input co-optimization, where multimodal inputs are provided per example, our goal is to optimize a single, reusable multimodal prompt that globally contextualizes an MLLM across all instances of a task.
>
> * Also, this shift introduces two challenges and methodological innovations that existing prompt optimization work does not address: (1) exploration in a cross-modal prompt space, where textual and non-textual components must be jointly updated while preserving semantic alignment, which motivates our alignment-preserving exploration strategy; and (2) selection in the greatly expanded multimodal prompt space, where good prompts are sparse, which motivates our prior-inherited Bayesian UCB mechanism that warm-starts evaluation to improve search stability and efficiency.
>
> We appreciate the opportunity to clarify these points and hope that our explanation makes the novelty of our formulation and methodology clearer.
>
> ---
>
> > W2. The analysis lacks stronger insights into why multimodal prompts help; results show improvement, but don’t probe interpretability, modality interactions, or failure cases.
>
> Thank you for raising this point. We would like to emphasize that our work includes both quantitative and qualitative analyses aimed at understanding why multimodal prompts help, and these analyses reveal clear benefits that go beyond simple performance gains.
>
> First, our hidden-state visualization (Figure 8) offers interpretable evidence that multimodal prompts induce a meaningful shift in MLLM representations. Specifically, text-only optimized prompts occupy a narrow semantic region, whereas MPO’s multimodal prompts move hidden representations into a distinct space. This indicates that the non-textual component contributes genuinely new information rather than acting as redundant decoration.
>
> Second, we qualitatively analyze the optimized prompts to examine how the textual and non-textual components interact and remain cross-modality aligned in the “Qualitative Result” paragraph of Section 4.2 with additional analyses in Appendix C.3. In particular, as shown in Table 16 from Appendix C.3, the multimodal prompts from MPO combine concise textual guidance with the corresponding visual evidence for grounding, which enables the model to reason with concrete visual cues (e.g., bill curvature, crest shape, feather structure). In contrast, the (text-only) SEE prompts in Table 15 must encode all visual cues linguistically (e.g., “Pay attention to specific visual markers such as bill shape, color, and the feather crest on the Crested Auklet…”) and therefore often produce vague, high-level instructions that lead to coarse reasoning and frequent misclassifications (e.g., “The overall body structure and size are consistent with the Rhinoceros Auklet.”).
>
> Finally, we have conducted a focus failure analysis (on the Auklet subtask) and observed that among instances where the optimized prompts from the (text-only) SEE fail, our MPO resolves 66.6% of them. This result further supports that the multimodal cues provide additional, complementary information that text-only optimization cannot express.

---

> > ### Author Response · Authors · 2025-11-21
> > **Response to Reviewer z13R (2/2)**
> >
> > > Q1. How much of the gain actually comes from the added non-textual modality rather than the optimization process itself? A controlled text-only ablation would clarify this.
> >
> > This is an excellent suggestion. To isolate the contribution of the non-textual modality from the optimization procedure itself, we have performed an additional ablation study with MPO under two restricted settings: (1) MPO w/o Image, where we optimize only the textual prompt using our search procedure, and (2) MPO w/ Fixed Image, where we generate a non-textual prompt in the first iteration and keep it fixed while optimizing only the text afterwards. As shown in Table A.1, MPO w/o Image achieves an average accuracy of 69.7%, which is comparable to the text-only baseline SEE (69.0%). This indicates that the optimization procedure alone (without the non-textual component) does not account for the significant performance gain. Meanwhile, MPO w/ Fixed Image reaches 71.6%, showing that a static visual cue provides some benefit, but it still substantially lags behind the full MPO (76.4%). This demonstrates that simply adding a modality is insufficient and that the substantial gains stem from the joint optimization of both modalities in a cross-modal aligned manner.
> >
> > Table A.1. Ablation study on the PlantVillage dataset separating the effects of multimodal inputs and the optimization process.
> >
> > | 　 | Apple | Corn | Grape | Potato | Avg. |
> > |:---:|:---:|:---:|:---:|:---:|:---:|
> > | SEE | 79.8 | 75.9 | 44.7 | 75.7 | 69.0 |
> > | MPO w/o Image | 75.4 | 74.8 | 53.4 | 75.0 | 69.7 |
> > | MPO w/ Fixed Image | 75.7 | 76.0 | 58.2 | 76.3 | 71.6 |
> > | MPO | 77.7 | 78.2 | 65.9 | 84.0 | **76.4** |
> >
> > ---
> >
> > > Q2. Does the alignment-preserving exploration recover from visual semantic drift?
> >
> > Thank you for the insightful question. We note that while alignment-preserving exploration is primarily designed to maintain cross-modal consistency during joint updates, it also makes the process resilient to semantic drift in practice. First, the joint update step (driven by a unified feedback signal across textual and non-textual components) discourages candidates whose textual and non-textual prompts diverge semantically. Moreover, even in the rare case that an incoherent prompt is generated, our prior-inherited Bayesian UCB selection strategy acts as a filter: it evaluates all candidates, naturally assigning a low score to any poorly performing collapsed prompts and then discarding them. In other words, this ensures only high-performing, well-aligned candidates are selected for the subsequent iteration.
> >
> > ---
> >
> > > Q3. Robustness of Prior-Inherited Bayesian UCB to prior strength and when correlation assumptions break.
> >
> > As shown in Figure 9, we empirically analyze the sensitivity of our prior-inherited Bayesian UCB to the prior strength, and observe that moderate prior strengths (5–10%) consistently yield the best performance. This is because it strikes a balance between leveraging the parent prior and incorporating new evidence from the child. Specifically, when the prior strength is too small (e.g., $S=0$), the evaluation behaves like a cold start and under-utilizes useful prior information. Conversely, large prior strengths (20–25%) overfit to the parent and can bias the evaluation toward mediocre parents. Furthermore, even when the correlation between the parent and the child is weak, the moderate-strength regime remains stable, since the evaluation rapidly incorporates new observations without over-relying on the prior. Therefore, a moderate prior strength keeps the procedure both flexible and robust.
> >
> > ---
> >
> > > Q4. How reproducible is this pipeline for researchers without access to certain models (GPT-based generators)?
> >
> > In Table 2 (Bottom Right) of the main paper, we analyze various generators, including open-source models. From this, we observe that even with a small open-source image generator like SANA1.5 (1.6B), our MPO (71.8) still clearly outperforms the strong text-only baseline SEE (69.0). This indicates that the gains primarily come from our MPO framework itself, not from privileged access to a particular model. In other words, the fact that MPO continues to work well with smaller, easily accessible generators suggests that the pipeline is fully reproducible for researchers without access to larger or proprietary models.

---

### Author Response · Authors · 2025-11-27

Dear Reviewers,

We greatly appreciate your time and efforts in reviewing our paper, as well as your insightful and constructive comments. We would like to inform you that we have diligently responded to each of your comments in great detail.

We fully understand that your time is extremely valuable and that you are likely managing many responsibilities. However, as the discussion window will be closed in the near future, we would be deeply grateful if you could kindly take a moment to review our response. Please let us know if you have any additional points, which would be immensely valuable to us.

Thank you once again for your service, and we look forward to your response!


Warm regards, Authors

---

### Author Response · Authors · 2025-12-03
**Summary of Response**

We would like to thank all reviewers for their time, effort, and constructive feedback, as well as the area chairs, senior area chairs, and program chairs for their consideration throughout the review process. As the review process does not allow continued discussions with reviewers, we wanted to offer a summary of our rebuttal to make clear how each concern has been addressed.

---

First, we propose Multimodal Prompt Optimization (MPO), which extends the prompt optimization space from text-only to discrete multimodal inputs, and we are deeply encouraged that the reviewers have recognized several strengths of our work:

- Rigorous Evaluation: The paper is supported by convincing and thorough experiments on multiple modalities, covering 10 benchmarks (z13R, oPXv, wMqy, DhB1). Also, analytical experiments on backbone models, ablation study, and qualitative analyses are well-designed and rigorous (z13R, oPXv, oxEi).

- Contribution & Significance: The proposed method is an interesting and promising direction (DhB1), revolutionizes the conventional prompt structure for MLLMs (wMqy), and fills a gap in the MLLM literature (z13R, oPXv).

- Soundness: The proposed method is technically sound, combining alignment-preserving exploration with prior-inherited Bayesian UCB to balance multimodal consistency and sample efficiency (oxEi).

- Presentation: The paper is well-written, well-structured, and easy to follow with clear motivation. (z13R, oxEi, wMqy, DhB1).

---

In addition to this, we have also faithfully addressed all the concerns with additional experiments and clarifications:

- Source of Gain (z13R, oxEi, DhB1): We have demonstrated that MPO outperforms strong text-only baselines even when using lightweight, open-source generators (e.g., SANA-1.5, 1.6B), confirming that gains stem from our optimization strategy, not just from powerful generator priors.

- Cost-Efficiency (oPXv, oxEi, wMqy): We have conducted a cost-constrained analysis showing MPO is highly efficient: MPO operating at a low budget (\$0.25) matches or exceeds state-of-the-art text-only methods operating at double the budget (0.50).

- Ablation & Mechanics (z13R, wMqy): We have provided ablations, demonstrating that gains arise specifically from the joint optimization of cross-modal alignment, rather than simply adding a modality or the optimization process alone.

- Novelty compared to multimodal conditioning and input co-optimization methods (z13R): We have clarified that the goal and settings are clearly different: unlike the suggested works, where the multimodal inputs are provided per query, we aim to optimize a single, reusable multimodal prompt that globally contextualizes an MLLM across all instances of a task. Moreover, we have emphasized that our work proposes a novel framework for multimodal prompt optimization by introducing an alignment-preserving exploration and a prior-inherited selection mechanism, designed to address the new challenges arising from this shift.


- Diverse Benchmarks (oPXv): We have explained that MPO is evaluated on extensive benchmarks across diverse modalities, focusing on assessing the core perceptual capabilities, dynamic visual input understanding, and cross-modal complex reasoning. We have further conducted an additional experiment on the Geometry3K benchmark, demonstrating MPO’s effectiveness on tasks requiring complex geometric reasoning.

We believe that these clarifications and additional experiments show how we have addressed the concerns from all reviewers and further highlight the value and applicability of MPO as a robust, cost-effective framework for multimodal prompt optimization for MLLMs.

Sincerely,

Authors

---

### Meta-Review · Area_Chair_8B1C · 2026-01-05

**Summary:**

The reviewers mainly raised concerns about the novelty and positioning of multimodal prompt optimization, as well as whether the paper provides sufficient analysis to explain why multimodal prompts are effective. Several reviewers questioned whether the observed gains truly come from multimodal optimization itself, rather than from stronger generators or additional computational overhead, and whether the method can generalize to more challenging settings, such as tasks with many categories, standard multimodal benchmarks, or complex reasoning problems. Other concerns focused on the clarity of certain design choices (e.g., exploration operators, parent prompts), the robustness of cross-modal alignment, and the lack of direct cost comparisons with text-only prompt optimization methods.

**Reviewer Concerns:**

Reviewer z13R：\
The authors conducted the requested controlled text-only ablation study (Q1), which successfully isolated the performance gains to the cross-modal aligned manner. The authors also clarified the conceptual novelty regarding task-level optimization (W1) and substantiated claims of interpretability with hidden-state visualizations and failure analysis (W2). Concerns regarding the sensitivity of the Bayesian prior (Q3) and reproducibility were addressed using existing sensitivity analyses and validation on an open-source image generator model. The robustness against semantic drift (Q2) was explained through the algorithm's mechanism, which appears theoretically sound, though empirical traces of drift recovery were not explicitly visualized.

Reviewer oPXv：\
The authors address concerns regarding benchmark coverage. They maintain that the existing benchmark suite adequately encompasses perception-like skills without requiring static foundational benchmarks (e.g., MME). However, concerning the omission of reasoning tasks, they present significant new experimental results based on Geometry3K.
Criticisms regarding the absence of deployment costs (W2) are effectively addressed through the introduction of cost-constrained analysis: even under stringent financial constraints, the proposed method remains highly efficient and outperforms text-only baselines.

Reviewer oxEi:\
The authors addressed the reviewer's technical and definitional concerns with precision. They committed to revising the manuscript to explicitly define "multimodal prompts" and the "parent prompt" mechanism. The concern regarding generator dependency was countered by pointing to ablation studies where the method remained effective using lightweight, open-source models. Crucially, the concern regarding efficiency measurement was resolved with a new budget-constrained analysis that accounts for real-world API costs. Theoretical questions regarding the Bayes-UCB assumptions (Q1) and alternative retrieval strategies (Q2) were clarified through detailed explanations and references to comparative results in the paper.

Reviewer wMqy:\
The authors effectively addressed the reviewer's concerns regarding implementation and baselines. They provided concrete performance metrics for the "naive" sequential optimization approach (W1), demonstrating that the proposed joint optimization yields superior results (76.4 vs 71.4). The initialization process for multimodal prompts was clarified (W2). Furthermore, the concern regarding computational efficiency (W3) was well-addressed by shifting the comparison to a cost-constrained budget analysis.

Reviewer DhB1:\
The authors address the concern about generalization (W1, Q2) by noting that their evaluation already includes high-cardinality tasks, such as Drive&Act with 34 classes, as well as open-ended QA datasets. They also clarify that the exploration operators are designed to be applied in a flexible manner rather than following a fixed order (Q1), and provide a reasonable explanation for this design choice. In addition, the concern about dependence on high-performance generators (Q3) is mitigated by experimental results showing consistent improvements even when smaller, open-source models are used.

**Reviewer Scores:**

Based on the author's response and the complete review discussion record, the majority of reviewers are likely to maintain their original scores or slightly increase them following the discussion.

---

### Decision · Program_Chairs · 2026-01-26

Accept (Poster)